# Adaptive Latent-Space Constraints in Personalized Federated Learning

**Sana Ayromlou**[*]
Vector Institute & Google
Toronto, Ontario, CA
ayromlous@gmail.com

**D. B. Emerson**
Vector Institute
Toronto, Ontario, CA
david.emerson@vectorinstitute.ai

## Abstract

Federated learning (FL) is an effective and widely used approach to training deep learning models on decentralized datasets held by distinct clients. FL also strengthens both security and privacy protections for training data. Common challenges associated with statistical heterogeneity between distributed datasets have spurred significant interest in personalized FL (pFL) methods, where models combine aspects of global learning with local modeling specific to each client's unique characteristics. This work investigates the efficacy of theoretically supported, adaptive MMD measures in pFL, primarily focusing on the Ditto framework, a state-of-the-art technique for distributed data heterogeneity. The use of such measures significantly improves model performance across a variety of tasks, especially those with pronounced feature heterogeneity. Additional experiments demonstrate that such measures are directly applicable to other pFL techniques and yield similar improvements across a number of datasets. Finally, the results motivate the use of constraints tailored to the various kinds of heterogeneity expected in FL systems.

## 1 Introduction

Federated learning (FL) has become an indispensable tool for training deep learning models on distributed datasets. Such a setting arises naturally in many scenarios where, for various reasons including privacy, security, and resource constraints, training data should or must reside in disparate locations. Federally trained models receive updates from a larger collection of training data than models trained on single data silos and, as such, often demonstrate better performance, generalizability, and robustness [34]. In a typical FL system, clients each hold a distinct dataset on which a model is to be collaboratively trained. A server is used to perform model aggregation of some kind, requiring transfer of model weights but never raw data. In each round, clients perform model training using their local dataset. After a period of local training, the server aggregates the individual models, or subsets thereof, and sends the aggregated information back to the clients for another round of training.

While data heterogeneity is challenging in standard model training, it is particularly pernicious in FL settings due to the increased prevalence of heterogeneity in distributed datasets [19, 40]. Many works have aimed to address the effects of data heterogeneity in FL since the original mechanism of FedAvg was introduced [30]. One branch of study considers techniques for robust global optimization, seeking to restrict or correct local divergence [14, 28, 20, 45]. However, such approaches, referred to here as global FL methods, typically train a single model for all clients. In many instances, system and statistical heterogeneity limit the existence of a shared model that performs well across all FL participants [16, 3]. This has led to the rise of personalized FL (pFL) methods, which federally train distinct models for each client. Specifically, this work focuses on improving cross-silo pFL methods, where clients represent a small number of reliable institutions with sufficient training resources [13].

---

[*]Corresponding author. Work done while at the Vector Institute.

39th Conference on Neural Information Processing Systems (NeurIPS 2025).

A common technique for addressing heterogeneity during federated training is the augmentation of each client's local loss with a penalty discouraging large deviations in model weights or representations from a reference model during training. In this work, we propose integrating optimizable MMD measures [9] into penalty-based pFL frameworks, modifying the Ditto and MR-MTL algorithms as representative examples [21, 24]. Specifically, Multi-Kernel MMD (MK-MMD) [10] and MMD-D [23] are applied to constrain divergence between models' latent representations. Ditto remains one of the best performing pFL methods available, especially in challenging heterogeneous settings. Several recent benchmarks show that it delivers state-of-the-art performance in many settings [3, 39, 29]. As such, it is used here as the primary pFL approach for experimentation.

The contributions of this work are three-fold. First, it proposes using theoretically supported statistical distance measures as penalty constraints for dealing with heterogeneity in pFL, moving beyond previously considered paired feature-based constraints. Second, it demonstrates that leveraging the adaptability of MK-MMD or MMD-D measures through iterative re-optimization provides notable performance improvements in settings with high feature heterogeneity where existing methods, such as Ditto, under-perform. Finally, experiments with either natural or controllable levels of label and feature heterogeneity highlight the strengths and weaknesses of the types of drift penalties investigated. While Ditto is the main focus, additional results show that the proposed methodology is directly applicable to other penalty-based optimization techniques in FL and provides similar benefits.

## 2    Related Work

The most common formulation of penalty-based constraints in FL discourages model weight drift from a set of reference weights in the $\ell^2$ norm. This has been applied successfully in global FL methods like FedProx [20] and pFL approaches such as Ditto [21] and MR-MTL [24]. Recent work, has modified this constraint to consider feature representations. Both MOON [18] and PerFCL [46] use contrastive losses to encourage feature representations to be close, or far, from some reference. In certain settings, MOON demonstrates marked improvements over FedProx as a global FL approach, partially motivating the investigation here. However, the contrastive losses used in MOON and PerFCL are not adaptive and require careful hyperparameter tuning. Furthermore, in [39], PerFCL underperformed contemporary pFL approaches, including Ditto.

As part of the work proposing Ditto, the authors experiment with two alternatives to the standard $\ell^2$ norm to measure weight divergence, symmetrized KL divergence [12] and Fisher-weighted squared distances [44]. Therein, no performance improvements are achieved. In addition, these alternative regularizers are still applied to quantify differences in model weights, rather than latent spaces. Finally, the measures are not trainable as part of the learning process.

Prototype-based pFL methods, such as FedProto [38], share some similarities with this work. They aim to constrain local representations tied to class labels, or prototypes, from drifting too far from their global counterpart. However, these approaches are generally only applicable to classification tasks by design. Prototypes are also most useful near the final layers of a model, where class separation becomes sharp. Finally, drift is measured with $\ell^2$ or $\ell^1$ norms rather than the adaptive measures proposed here. There is, however, an opportunity for future work to explore the utility of integrating adaptive MMD measures into these techniques. Appendix E provides additional discussion of other, less related, pFL methodologies and how the modifications proposed here might be applied.

A few works have considered integrating MMD into FL systems. FedMMD incorporates a fixed MMD measure on local features as a means of modifying server-side aggregation in FedAvg [11]. This is strictly a global FL approach and applies MMD to modify model aggregation rather than local learning. In [42], a static MMD measure is constructed and used to constrain a model's latent space during client-side training. However, as with FedMMD, the proposed algorithm is not a pFL approach and does not consider kernel optimization. It also lacks robust experimentation. Non-adaptive MMD measures have also been used successfully to overcome feature-distribution shifts when federally training generative models for local data augmentation [4]. Finally, various forms of MMD have been used in other areas, most widely in hypothesis testing and domain adaptation [10, 24, 25, 41, 17]. To our knowledge, this is the first study considering the utility of substituting or augmenting the loss regularizers of Ditto and MR-MTL with an adaptive, feature-based regularizer in the forms of MK-MMD and MMD-D. Because weight-based, $\ell^2$ penalties are common in other global and pFL methods, the results have broader implications beyond improving any single pFL technique.

# 3 Methodology

At its core, FL enables multiple clients to train models collaboratively on distributed data while sharing only model parameters with a server, rather than transferring raw data. Throughout, assume there are $N$ clients with datasets, $D_1, \ldots, D_N$ such that $D_i = \{(x_j, y_j)\}_{j=1}^{n_i}$. Let $n = n_1 + \ldots + n_N$. Each client incorporates a local loss function $\ell_i$ parameterized by some set of weights $w$. In the sections to follow, the Ditto algorithm is used as an exemplary approach for experimentation. However, the methodology is transferrable to other pFL methods, with additional experiments showing similar benefits for MR-MTL through the proposed techniques.

Ditto maintains two models, sharing the same architecture, on each client $i$, a global model with weights $w_G^{(i)}$ and a local model with weights $w_L^{(i)}$. Algorithm 1 summarizes Ditto training, where $T$ denotes the number of FL rounds, $s$ denotes the number of local training steps performed by each client, and $\bar{w}$ is the initial set of weights for both the global and local models. In practice, any optimizers may be used to train the global and local models, with potentially different learning rates. Likewise, the number of local training steps may differ. Here, however, as in the original work, these are coupled for simplicity. For standard Ditto, the measure $d(w_L^{(i)}, \bar{w}) = \|w_L^{(i)} - \bar{w}\|_2^2$.

---

**Algorithm 1:** Ditto algorithm with FedAvg aggregation and batch SGD for local optimization.

---

**Input:** $N, T, s, \lambda, \eta, \bar{w}$.
Set $w_L^{(i)} = \bar{w}$ for each client $i$.
**for** $t = 0, \ldots, T-1$ **do**
    **for** *each client $i$ in parallel* **do**
        Set $w_G^{(i)} = \bar{w}$.
        **for** *$s$ iterations, draw batch $b$* **do**
            $w_G^{(i)} = w_G^{(i)} - \eta \nabla \ell_i \left( b; w_G^{(i)} \right)$.
            $w_L^{(i)} = w_L^{(i)} - \eta \nabla \left( \ell_i \left( b; w_L^{(i)} \right) + \frac{\lambda}{2} d(w_L^{(i)}, \bar{w}) \right)$.
        **end**
        Send $w_G^{(i)}$ to server for aggregation.
    **end**
    $\bar{w} = \frac{1}{n} \sum_{i=1}^{N} n_i \cdot w_G^{(i)}$.
**end**

---

## 3.1 Beyond Weights: Feature-Drift Constraints

The $\ell^2$-weight constraint imposed by Ditto has been effective in many settings. However, the relationship between the type and degree of data heterogeneity to the most effective kind of drift-constraint remains under-explored. In this work, we experiment with both weight- and feature-drift penalties, analyzing their impact on learning personalized models in various settings. To define the adaptive constraints investigated below, consider splitting a model into one or several stages, with intermediate outputs representing latent features. For simplicity of presentation, the setting of a single latent space is discussed here, where the extension to additional layers is straightforward. The global and local models, along with their weights, $w_G = [\theta_G, \phi_G]$ and $w_L = [\theta_L, \phi_L]$, are decomposed into a feature extractor $f(\cdot; \theta)$ and a classifier $g(\cdot; \phi)$. For an input $x$, local predictions are generated as $\hat{y} = g(f(x; \theta_L); \phi_L)$. Global predictions are produced analogously. In Algorithm 1, the reference weights, $\bar{w}$, are also decomposed as $\bar{w} = [\bar{\theta}, \bar{\phi}]$.

Rather than penalizing divergence of the local model from the global one via the $\ell^2$ distance between weights, a measure of drift between latent features is defined. Such constraints may be constructed in two ways: paired or unpaired. In paired approaches, which are widely used [18, 46], for each input $x$, the aim is to minimize $d(f(x; \theta_L), f(x; \theta_G))$. This enforces point-wise alignment, but disregards statistical interpretations of the feature space. In contrast, we propose an unpaired approach which reduces the distance between the probability distributions of the global and local latent spaces, allowing greater flexibility for models to adapt based on data heterogeneity. Here, cosine similarity is used as a baseline to represent paired feature alignment by maximizing similarity.

## 3.2 Maximum Mean Discrepancy

Let $X$ be some topological space and $P$ and $Q$ be Borel probability measures on $X$. Any symmetric and positive-definite kernel $k : X \times X \to \mathbb{R}$ induces a unique Reproducing Kernel Hilbert Space (RKHS), $\mathcal{H}_k$, into which $P$ and $Q$ are embedded and their distance measured [33]. This is written

$$\text{MMD}^2(P, Q; \mathcal{H}_k) = \mathbb{E}_{(x,x') \sim (P,P)} \, k(x, x') + \mathbb{E}_{(y,y') \sim (Q,Q)} \, k(y, y') - 2\mathbb{E}_{(x,y) \sim (P,Q)} \, k(x, y).$$

In this work, $X \subseteq \mathbb{R}^m$, where $m$ is a model's latent-space dimension, and the feature extraction modules $f(x; \theta_L)$ and $f(x; \bar{\theta})$ produce distributions on $X$.

MMD measures are a powerful tool for efficiently measuring the distance between two distributions. However, their effectiveness depends heavily on the kernel, $k$. As such, kernel optimization techniques are crucial. Herein, the primary goal of kernel selection is to maximize test power, i.e. the ability of the measure to accurately distinguish two distributions. Two kernel optimization techniques are considered to produce strong measures of feature-space discrepancies: MK-MMD and MMD-D. The former optimizes a linearly weighted combination of radial basis functions (RBFs) of varying length scale. The latter leverages a learnable deep kernel to construct a strong distance metric. RBFs are not the only kernel type used in MMD, but they are a common and high-performing choice [35].

## 3.3 Multi-Kernel MMD

In [10], a set of possible kernels is defined as

$$\mathcal{K} = \left\{ k \mid k = \sum_{j=1}^{d} \beta_j k_j, \sum_{j=1}^{d} \beta_j = 1, \beta_j \geq 0, \forall j \in \{1, \ldots, d\} \right\},$$

where $\{k_j\}_{j=1}^{d}$ is a set of symmetric and positive-definite functions, $k_j : X \times X \to \mathbb{R}$. Note that any $k \in \mathcal{K}$ uniquely defines an RKHS and associated measure

$$\text{MK-MMD}^2(P, Q; \mathcal{H}_k) = \sum_{j=1}^{d} \beta_j \text{MMD}^2(P, Q; \mathcal{H}_{k_j}).$$

Intuitively, any $k \in \mathcal{K}$ produces a measure of discrepancy between two distributions. The novel idea of [10] is to optimize the coefficients, $\beta_j$, to minimize the Type-II error in testing if two distributions, $P$ and $Q$, are the same for a fixed Type-I error. That is, one engineers an RKHS that has the best chance of properly detecting when two distributions differ through specification of an optimal kernel.

Denoting by $\beta \in \mathbb{R}^d$ the coefficients of the kernels $k_j$, Type-II error minimization is expressed as

$$\beta_* = \arg\max_{\beta \geq \mathbf{0}} \frac{\text{MK-MMD}^2(P, Q; \mathcal{H}_k)}{\sigma(P, Q, \mathcal{H}_k)}, \tag{1}$$

where $\sigma^2(P, Q, \mathcal{H}_k)$ is the variance of $\text{MK-MMD}^2(P, Q; \mathcal{H}_k)$. This variance has the form $\sigma^2(P, Q, \mathcal{H}_k) = \beta^T Q_k \beta$, where $Q_k$ is the $d \times d$ covariance matrix between kernels, $k_j$, with respect to $P$ and $Q$.

Let $\hat{X}$ and $\hat{Y}$ be sets of samples drawn from $P$ and $Q$, respectively. To approximately solve Equation (1), empirical estimates of each $\text{MMD}^2(P, Q; \mathcal{H}_{k_j})$, gathered into a vector $\hat{m} = [\hat{m}_1, \ldots, \hat{m}_d]^T$, and $Q_k$, denoted $\hat{Q}_k$, are computed. Following [10], the objective is approximated and rewritten as the quadratic program

$$\hat{\beta}_* = \arg\min_{\substack{\beta^T \hat{m} = 1 \\ \beta \geq \mathbf{0}}} \beta^T (\hat{Q}_k + \epsilon I)\beta, \tag{2}$$

where $\epsilon > 0$ is a stabilizing shift and $I$ is the $d \times d$ identity matrix.

In the experiments, $\epsilon = 1\text{e-}3$. The kernels selected to form $\mathcal{K}$ are RBFs of the form $k_j(x, y) = e^{-\gamma_j \|x-y\|_2^2}$ for a set of $\{\gamma_j\}_{j=1}^{d}$. Note that this optimization is only valid if $\hat{m}$ contains at least one positive entry. If this is not the case, [10] suggests selecting the kernel, $k_j$, that maximizes the ratio $\text{MMD}^2(P, Q; \mathcal{H}_{k_j})/\sigma(P, Q; \mathcal{H}_{k_j})$. Finally, linearly scaling estimates for both $\text{MK-MMD}^2(P, Q; \mathcal{H}_k)$ and $Q_k$ are provided in [10]. Unfortunately, these estimates performed poorly empirically. As such, they are not used in the experiments to follow.

### 3.4 MMD-D

Some studies have shown that simple RBF kernels, varying only in length-scale, $\gamma$, may underperform in more complex spaces where distributions vary significantly across regions [23, 35]. To address this, the use of deep kernels, which can adapt to the diverse structures needed in different regions, has been proposed. Here, the structure of [23] is used. Given inputs $x$ and $y$, a deep kernel is constructed by applying a kernel function, $k$, to the output of a featurization network, $\varphi$, parameterized by $\omega$ as $k(\varphi(x;\omega), \varphi(y;\omega))$. Alone, such a kernel may inadvertently learn to treat distant inputs as overly similar. To mitigate this, a safeguard is introduced defining the full deep kernel as

$$k_\omega(x, y) = (1 - \epsilon)k(\varphi(x;\omega), \varphi(y;\omega)) + \epsilon q(x, y), \tag{3}$$

where $q$ is a separate kernel acting on the unmodified input.

As in Section 3.3, RBFs are used such that $k(x,y) = e^{-\gamma_k \|x-y\|_2^2}$ and $q(x,y) = e^{-\gamma_q \|x-y\|_2^2}$. The variables of $\epsilon$, $\gamma_k$, $\gamma_q$, and $\omega$ are optimized monolithically. Again, the goal of such optimization is to minimize the Type-II error of the induced measure which, when using the kernel in Equation (3), is denoted MMD-D$^2(P, Q; \mathcal{H}_k)$. Similar to MK-MMD, this error minimization takes the form

$$\max_{\omega, \epsilon, \gamma_k, \gamma_q} \frac{\text{MMD-D}^2(P, Q; \mathcal{H}_k)}{\sigma(P, Q; \mathcal{H}_k)}, \tag{4}$$

where $\sigma^2(P, Q; \mathcal{H}_k)$ is now the variance of MMD-D$^2(P, Q; \mathcal{H}_k)$.

For samples $\hat{X}$ and $\hat{Y}$ drawn from $P$ and $Q$, respectively, the estimate for MMD-D$^2(P, Q; \mathcal{H}_k)$ derived in [23, Equation 2] is used. To estimate $\sigma^2(P, Q; \mathcal{H}_k)$, the regularized estimator of [23, Equation 5] is applied with $\lambda = 1e\text{-}8$. A fixed number of steps using an AdamW optimizer [26] with a learning rate of 1e-3 is applied to approximately solve Equation (4) and learn kernel $k_\omega(x, y)$.

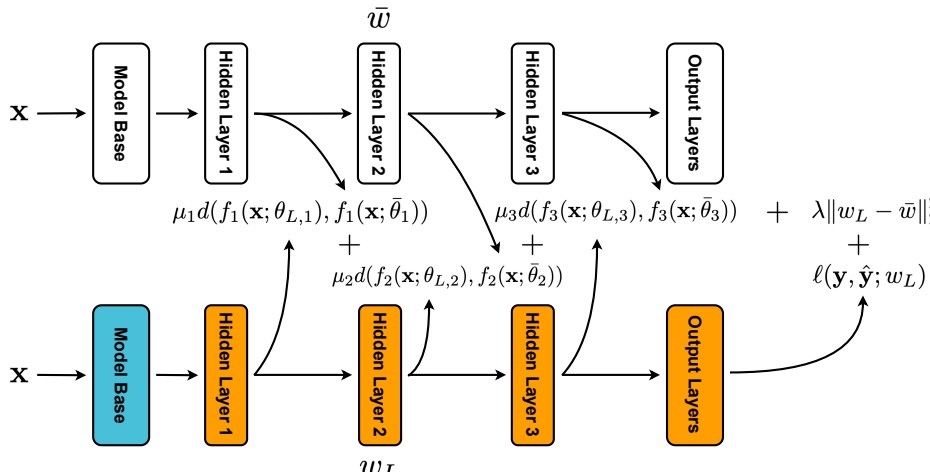

Figure 1: Feature-drift constraints applied to three latent spaces of a model, where $(\mathbf{x}, \mathbf{y})$ represent a batch of labeled data. In Ditto, the frozen global model (top) is used to constrain the local model (bottom) during client-side training.

### 3.5 Adaptive MMD Measures in FL

Provided feature-extraction maps for the global and local models in the Ditto framework, $f_i(x; \bar{\theta}_i)$ and $f_i(x; \theta_{L,i})$, the MMD measures defined in the previous section are used to augment the local model's loss function during training with weights $\mu_i$. This process is exhibited in Figure 1 for three intermediate latent spaces. Note that $(\mathbf{x}, \mathbf{y})$ in the figure represents batches of data rather than individual data points. These measures may be used in isolation or combined with the standard Ditto penalty of $\lambda \|w_L^{(i)} - \bar{w}\|_2^2$. Both MK-MMD and MMD-D aim to adapt non-linear kernels to measure the distance between two distributions more accurately.

In a static setting, where $P$ and $Q$ are fixed, the optimization processes for these approaches yield strong measures to distinguish these distributions. Here, these techniques are used to measure and constrain the differences in distributions of model latent spaces which evolve significantly throughout the FL training process. As the model feature maps evolve, previously optimal kernels become sub-optimal. To address this, we propose taking advantage of the optimization procedures discussed above to periodically re-optimize the MMD measures using samples of training data. In this way, strong measures of latent-space divergence are maintained throughout training.

## 4    Datasets

To quantify the utility of the proposed measures, four datasets are used. Each poses unique challenges in the FL setting. The first set of experiments considers several variants of CIFAR-10 with differing levels of label heterogeneity. This heterogeneity is synthetically induced using Dirichlet allocation, a common strategy in previous work [7, 2, 3]. Smaller values of the allocation parameter, $\alpha$, result in more heterogeneous clients. Values of $\{5.0, 0.5, 0.1\}$ are used to split data between five clients. The second dataset, referred to as Synthetic, is a generated dataset with controllable levels of feature heterogeneity across eight clients. It is an extension of the approach used in [20]. Heterogeneity depends on parameters $\alpha$ and $\beta$, with larger values corresponding to more heterogeneity. Two datasets are generated with $\alpha = \beta = 0.0$ and $\alpha = \beta = 0.5$. Inputs have 60 numerical features mapped to one of 10 possible classes. Appendix A provides details on the generation process.

The final two datasets focus on real clinical tasks with natural client splits and heterogeneity. Fed-ISIC2019 is drawn from the FLamby benchmark [8] and consists of 2D dermatological images to be classified into one of eight melanoma categories. The data is split across six clients. The RxRx1 dataset [36] is composed of 6-channel fluorescent microscopy images, each highlighting different cellular organelles. The task is to classify which genetic treatment, if any, the cells received based on the image. The original dataset contains 125K+ images across 1,108 classes. Following [4], only samples corresponding to the 50 most common classes are selected and three of the image channels are used. The data is partitioned into four clients based on the hospitals where the images were collected. This dataset exhibits significant feature heterogeneity. Both Fed-ISIC2019 and RxRx1 are quite challenging for global FL methods like FedAvg.

For all tasks, performance is measured through standard accuracy, with the exception of Fed-ISIC2019. Because the labels in Fed-ISIC2019 are highly imbalanced, balanced accuracy is used to better quantify model performance and matches the metric used in the FLamby benchmark.

## 5    Experimental Setting

In the experiments, hyperparameter sweeps are conducted to calibrate items such as learning rate for all methods. A full list of the hyperparameters considered and their optimal values appears in Appendix B. Other parameters, such as batch size, are detailed in Appendix C. The metrics reported in the results are the average of values across three training runs. Accompanying standard deviations are gathered in Appendix G. Within each FL training run, the final metric is the uniform average of each client's performance on their respective test sets.[2]

Each task uses a different model architecture. For the CIFAR-10 datasets, a CNN is used with two convolutional layers, each of which are followed by batch normalization (BN) and max pooling. The final two stages are fully connected (FC) layers ending in classification, matching the benchmark in [3]. Experiments are conducted with up to three latent spaces constrained with the MMD-based drift penalties. These are the output of the first FC layer, the output of the second BN layer, and the output of the first BN layer. When referring to the constraint "depth," 1 corresponds to constraining only the FC layer, 2 corresponds to adding the second BN layer, and 3 refers to adding the first BN layer (e.g. Figure 1). During experimentation with CIFAR-10, the optimal "depth" for the MMD constraints was uniformly 1. Because adding additional constraints did not improve performance, the remaining experiments simply applied MMD-based constraints at a single latent layer. For a more detailed discussion and results, see Section 6.2.

---

[2]All code is found at: `https://github.com/VectorInstitute/FL4Health/tree/main/research`

The model for the Synthetic datasets is a simple two-layer DNN followed by a softmax layer. The first layer serves as the feature extractor for constraint purposes. The second layer produces classification. For Fed-ISIC2019, EfficientNet-B0 [37], pretrained on ImageNet, is fine-tuned. The final linear layer serves as the classifier, and the rest of the model functions as the feature extractor. This architecture is also used in the FLamby benchmark [8]. Finally, a pretrained ResNet-18 model is fine-tuned for the RxRx1 task, as done in [4]. The MMD penalties are applied to the output of the model prior to the final classification layer.

During local training, two ways of updating the MMD measures are considered. In the first approach, the measures are updated every 20 steps of model training using $z$ batches of sampled training data. Alternatively, the measures are updated after every training step using the same batch of data. In either setting, the MMD-D kernel is optimized using AdamW. For MK-MMD, both approaches are reported and denoted with a subscript 20 or -1, respectively. For the CIFAR-10 and Synthetic datasets, MMD-D is trained using the periodic update approach with 5 optimization steps, while on Fed-ISIC2019 and RxRx1, it is updated after every training step with 25 optimization steps.

## 5.1 Communication, Privacy, and Computation Implications

With the proposed modifications to Algorithm 1, communication costs between clients and the server remain the same as standard Ditto. Only the global model is exchanged for aggregation and feature representations remain local. As such, the privacy properties are also nearly identical to Ditto, except that local training data is periodically used to optimize the MMD kernels on each client. There is also a small uptick in memory overhead with the need to store a modest number of feature activations and to train a set of kernel weights for MMD-D. The largest impact associated with the proposed approach is the computational cost of adapting the MMD measures during training. Optimizing the kernels for a batch of training data adds $\mathcal{O}(m^{3.5})$ and $\mathcal{O}(kq)$ arithmetic operations to each local training step for MK-MMD and MMD-D, respectively, where $m$ is the latent-space dimension, $k$ is the number of kernel optimization steps, and $q$ is the arithmetic operations associated with an SGD step for $\varphi$. For more details on computational costs and runtime complexity, see Appendix D.

## 6 Results

In the first set of experiments, performance of the MMD-based feature constraints is considered in isolation. That is, the traditional Ditto constraint in Algorithm 1 is replaced with one of the proposed adaptive measures. As a baseline, training with a cosine-similarity measure to penalize feature dissimilarity is also explored. The results are reported in Table 1.

Across all datasets, and heterogeneity levels, at least one of the MMD-based penalties outperforms the cosine-similarity baseline. In the case of the Synthetic, Fed-ISIC2019, and RxRx1 datasets, the performance gaps are fairly large. These results validate the effectiveness of the proposed measures in guiding federated training by adaptively limiting feature-representation drift. In most cases, using the feature-based penalties within the Ditto framework significantly outperforms FedAvg. The only setting where this is not the case is CIFAR-10 with $\alpha = 5.0$. This is the most homogeneous setting, and the convergence of FedAvg is minimally impacted. Below, combining the standard Ditto penalty with MMD-based measures boosts performance past FedAvg, even for $\alpha = 5.0$.

For the second set of experiments, the MMD-based measures are combined with the original Ditto penalty to consider the complementary utility of applying both constraints. The resulting local model loss function for client $i$ is written

$$\ell_i\left(b; w_L^{(i)}\right) + \frac{\lambda}{2}\|w_L^{(i)} - \bar{w}\|_2^2 + \mu d\left(f_i(b; \theta_L^{(i)}), f_i(b; \bar{\theta})\right), \tag{5}$$

where $b$ represents a batch of training data and $d(\cdot, \cdot)$ denotes one of the MMD-based measures. For comparison, standard Ditto is also used to federally train models for each task. This is equivalent to setting $\mu = 0$ in Equation (5). The results for each dataset are summarized in Table 2.

Comparing the results of Tables 1 and 2, vanilla Ditto outperforms both MMD-D and MK-MMD when used in isolation for all variants of CIFAR-10 and Fed-ISIC2019. However, for the Synthetic and RxRx1 datasets, which demonstrate notable feature heterogeneity, using either MMD-D or MK-MMD yields large performance improvements compared to standard Ditto. For RxRx1, including the Ditto penalty produces a small, additional improvement with the MMD-D measure, but can

Table 1: Average performance when **replacing** the standard Ditto constraint with various feature-drift penalties. Bold and underline indicate the best and second best value across **pFL methods**. Subscripts for CIFAR-10 and Synthetic indicate values for $\alpha$ and $\alpha = \beta$, respectively.

| Dataset | FedAvg | Ditto | Feature Drift Constraint | | | |
| --- | --- | --- | --- | --- | --- | --- |
| | | | Cos. Sim. | MMD-D | MK-MMD$_{-1}$ | MK-MMD$_{20}$ |
| CIFAR-10$_{0.1}$ | 71.220 | **84.930** | 84.212 | 83.789 | 84.136 | 84.439 |
| CIFAR-10$_{0.5}$ | 75.575 | **80.702** | 75.167 | 75.094 | 75.678 | 76.564 |
| CIFAR-10$_{5.0}$ | 77.284 | **77.658** | 67.298 | 67.729 | 68.718 | 68.832 |
| Synthetic$_{0.0}$ | 84.733 | 89.129 | 89.975 | 90.237 | **91.418** | 90.066 |
| Synthetic$_{0.5}$ | 85.458 | 85.533 | 90.199 | **91.270** | 91.137 | 90.262 |
| Fed-ISIC2019 | 64.057 | **71.350** | 61.269 | 64.302 | 60.168 | 62.677 |
| RxRx1 | 35.207 | 65.629 | 64.985 | **67.478** | 65.861 | 67.078 |

Table 2: Average performance when **augmenting** the standard Ditto constraint with various feature drift constraints. Bold and underline indicate the best and second best value across methods. Subscripts for CIFAR-10 and Synthetic indicate values for $\alpha$ and $\alpha = \beta$, respectively.

| Dataset | FedAvg | Ditto | + Feature Drift Constraint | | | |
| --- | --- | --- | --- | --- | --- | --- |
| | | | Cos. Sim. | MMD-D | MK-MMD$_{-1}$ | MK-MMD$_{20}$ |
| CIFAR-10$_{0.1}$ | 71.220 | 84.930 | 84.924 | **85.214** | 84.723 | 84.900 |
| CIFAR-10$_{0.5}$ | 75.575 | 80.702 | 80.669 | 80.696 | 80.936 | **80.976** |
| CIFAR-10$_{5.0}$ | 77.284 | 77.658 | **78.052** | 77.739 | 77.578 | 77.739 |
| Synthetic$_{0.0}$ | 84.733 | 89.129 | 89.187 | **89.458** | 89.183 | 89.258 |
| Synthetic$_{0.5}$ | 85.458 | 85.533 | 87.783 | **89.695** | 88.154 | 88.104 |
| Fed-ISIC2019 | 64.057 | 71.350 | 70.970 | **72.226** | 71.169 | 71.267 |
| RxRx1 | 35.207 | 65.629 | 67.027 | **67.755** | 65.984 | 66.892 |

degrade accuracy with MK-MMD. Similarly, adding the Ditto penalty for the Synthetic datasets hurts performance across the board. By construction, the Synthetic datasets have aspects that are adversarial to weight-based constraints, as the feature and label generation process incorporates normally distributed local weight variations. As such, these constraints may hinder learning at a certain point. As noted by the dataset creators, RxRx1 exhibits deep feature heterogeneity, which is especially challenging for FL algorithms. The results suggests that the proposed MMD-based measures are particularly well-suited for the kind of underlying heterogeneity in these datasets, while standard Ditto constraints are less effective.

The results in Table 2 also show that combining the traditional Ditto constraint with an MMD-based one produces the best results for Fed-ISIC2019 and CIFAR-10, in all but one setting. In the case of CIFAR-10, the improvements are modest but persistent across different levels of heterogeneity. Interestingly, despite yielding less favorable results as a standalone constraint, pairing the Ditto and cosine similarity penalties does provide the best performance for the most homogeneous version of CIFAR-10. In this setting, the MMD-based constraints remain second best. When using MMD-D in combination with Ditto for Fed-ISIC2019, the improvement is more prominent, especially given how challenging the task is [8, 39]. Finally, the combined results imply that the flexibility offered by the deep kernel of MMD-D produces better results than MK-MMD, especially for complex datasets.

The results above demonstrate that replacing or augmenting the traditional Ditto penalty with the adaptive MK-MMD or MMD-D constraints yields performance benefits. Notably, the proposed approach is transferrable to other pFL methods that use penalties to guide local training. To exhibit the utility of the adaptive constraints in other settings, experiments augmenting MR-MTL, another high-performing pFL method, with MK-MMD or MMD-D as latent-space penalties are conducted. The results are reported in Appendix F. Integrating the adaptive measures produces accuracy improvements over the standard MR-MTL algorithm in all settings, with the exception of FedISIC-2019.

## 6.1 Interaction of the Ditto and MMD Constraints

For the Synthetic and RxRx1 tasks, the traditional Ditto penalty under-performs compared to applying either MK-MMD or MMD-D in isolation. However, for the other datasets, a natural question arising from the results is whether the benefits of adding the MMD-based penalties are captured by simply increasing the Ditto constraint weight. To consider this, performance is measured for a variety of $\lambda$ near the optimal value for the CIFAR-10 and Fed-ISIC2019 datasets. Ditto is both augmented with the MMD-based measures and considered alone. The results are shown in Figure 2.

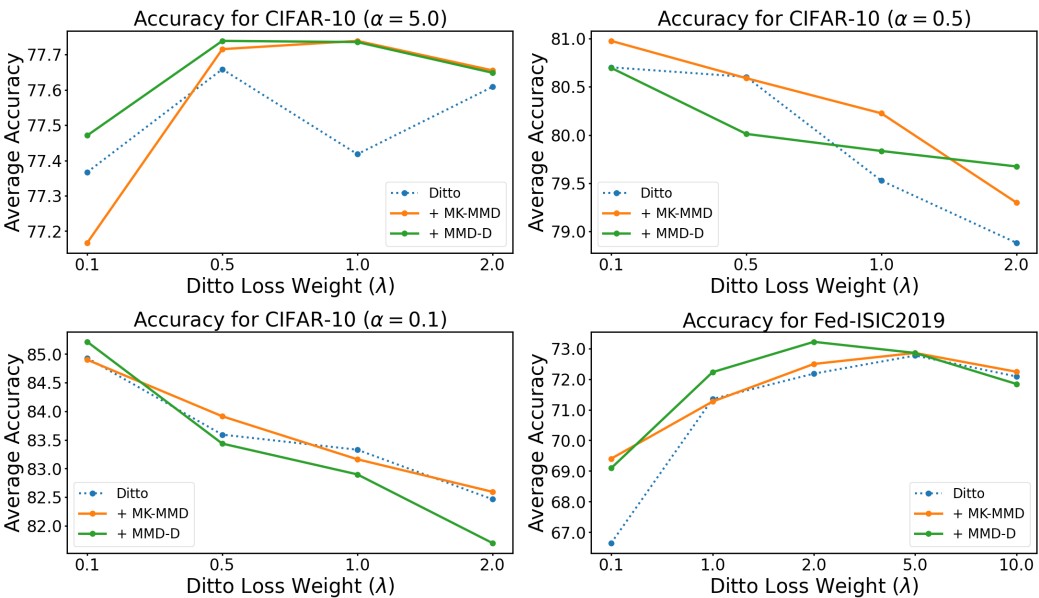

Figure 2: Results for CIFAR-10 and Fed-ISIC2019 when varying $\lambda$ around the optimal value. The best result is reported for MMD-based penalty weights, $\mu$, drawn from $\{0.01, 0.1, 1.0\}$.

For $\alpha = 5.0$, adding MMD-D outperforms Ditto-only for all values of $\lambda$. MK-MMD also improves performance for $\lambda$ at or above the best value. In the $\alpha = 0.5$ setting, augmentation with MK-MMD is better or equivalent to standard Ditto with increasing $\lambda$. In this setting, MMD-D under-performs for $\lambda = 0.5$ but is equivalent or better than Ditto for other values. In the high-heterogeneity regime, the results are more mixed. For the optimum $\lambda$, adding MMD-D provides a performance boost, but decays more rapidly than Ditto as $\lambda$ increases. The addition of MK-MMD simply yields comparable performance. At the highest levels of heterogeneity, weaker constraints, which permit more substantial local model divergence, can be advantageous. As such, the stronger penalties provided by augmentation with the MMD-based measures may be less helpful. For Fed-ISIC2019, both MMD-D and MK-MMD maintain an advantage over Ditto for nearly all values of $\lambda$. MMD-D achieves the highest peak value by a good margin.

Overall, the results demonstrate that the benefits of the new constraints are not simply replicated by increasing $\lambda$. Rather, these constraints capture beneficial aspects of drift not considered by the purely weight-based penalty. Intuitively, the weight- and feature-based constraints impact training dynamics differently. Weight-drift penalties weakly constrain a model to a neighborhood of reference weights. However, with model depth and nonlinearity, small weight perturbations can still produce large deviations in representations and outputs. We conjecture that combining such penalties with strong feature constraints can produce a more uniform drift penalty, providing complementary benefits in certain settings. Furthermore, in many contexts, a single form of heterogeneity will not characterize all drift between FL clients. Thus, penalizing drift in distinct ways is likely to be advantageous.

## 6.2 Ablation Studies

In the hyperparameter sweeps for CIFAR-10, the best results are obtained with a constraint depth of 1. This is illustrated on the left of Figure 3. For a fixed $\lambda = 0.1$, accuracy degrades with the

introduction of additional penalties. We conjecture that constraining too many layers in a relatively small model, as used in the CIFAR-10 experiments, produces a heavy constraint and hinders learning. This relationship informed the decision to apply MMD-based constraints to the latent space of a single layer for the remaining datasets. Further performance improvements may indeed exist for other datasets with different constraint configurations, as this simply limited the configuration search space.

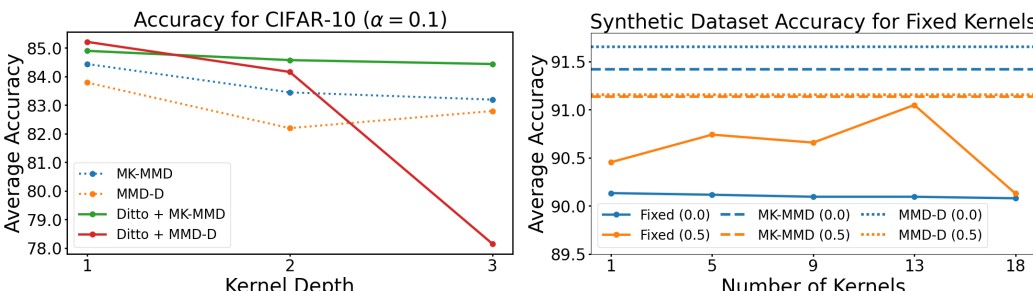

Figure 3: Evolution of accuracy on the CIFAR-10 ($\alpha = 0.1$) dataset (left) with an increasing number of constraints at various model depths for the MMD measures with and without the standard Ditto drift penalty. Accuracy for Synthetic datasets (right) using a varying number of fixed kernels with uniform weight compared to using the kernel optimization procedures of MK-MMD or MMD-D.

Improvements to test power when optimizing MMD kernels has been established in previous work [23, 35]. However, to validate that the kernel optimization procedures translate to task improvements, the performance of these approaches is compared to using a fixed set of RBF kernels linearly combined with uniform weights. In the experiment, the first $r$ kernels are selected from the original 18 generated by $\gamma_j$ values from $2^{-3.5}$ to $2^1$ with exponents evenly spaced in increments of $0.25$. For example, when $r = 5$, kernels corresponding to $\gamma_j = 2^{-3.5}, 2^{-3.25}, 2^{-3}, 2^{-2.75}, 2^{-2.5}$ are used. Each receives a uniform weight equal to $1/r$ such that the MMD measure is the average value of the individual measures induced by the fixed kernels. Results for the two Synthetic datasets are shown on the right in Figure 3. Kernel optimization with MK-MMD and MMD-D both outperform all fixed kernel settings. Notably, the 18-kernel case is equivalent to the MK-MMD setting in previous experiments, but with the kernel weight optimization procedure disabled. The optimization processes clearly generate substantive improvement in accuracy in this setting.

## 7 Conclusions, Limitations, and Future Work

In this work, two adaptive MMD-based measures are proposed to constrain feature representation divergence between models in FL. The Ditto and MR-MTL frameworks for pFL are expanded to incorporate such measures, penalizing local-model drift from a model aggregated with FedAvg on the server. We find that using these measures of feature-space distributions alone, or in tandem with traditional weight-based distances, produces marked performance improvements across several tasks, including important real-world, clinical benchmarks. The main results and ablation studies demonstrate that, rather than simply providing a parallel penalty to the standard weight-drift constraints, the MMD-based measures differ in kind. These measures appear especially effective in settings with significant feature heterogeneity between clients, such as in the Synthetic and RxRx1 datasets. Both measures are effective, but MMD-D empirically provides more consistent improvements.

While the results of this work are promising, some limitations are worth noting. The experiments focus on the utility of the MMD measures in the context of pFL frameworks with weight-based penalties. However, other methods, such as FedProto, may also benefit from introducing such penalties. Moreover, while the experiments indicate settings where the proposed measures offer notable benefits, this work does not theoretically quantify the conditions under which such measures are most effective. Some preliminary thoughts on such theory are provided in Appendix H, but more work is required. Finally, as noted in Section 5.1, the kernel optimization procedures introduce non-trivial computational costs. Avenues for reducing such costs are discussed in Appendix D. The above questions and limitations will be the subject of future work.

## Acknowledgments

The authors would like to extend a heartfelt thank you to Fatemeh Tavakoli of the Vector Institute for her help during the rebuttal period. Her efforts made this publication possible and we are truly grateful. We would also like to thank the anonymous referees for their helpful comments and suggestions that improved the quality of this work.

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

# A  Synthetic Dataset Generation Details

Label heterogeneity of varying levels of intensity can be imposed on public datasets using Dirichlet allocation. Inducing controllable levels of feature heterogeneity is more challenging. To simulate feature-level heterogeneity, we extend the approach outlined in [20]. Therein, to generate data for $k$ clients, samples $(X_k, Y_k)$ are synthesized using a simple one-layer network as $y = \arg\max(\text{softmax}(Wx + b))$. To produce a more complicated and multi-layer relationship, the generative function is extended to use a two-layer network as

$$y = \arg\max\left(\text{softmax}\left(W_2\left(T^{-1}(W_1 x + b_1)\right) + b_2\right)\right),$$

where $x \in \mathbb{R}^{60}$, $W_1 \in \mathbb{R}^{20\times60}$, $b_1 \in \mathbb{R}^{20}$, $W_2 \in \mathbb{R}^{10\times20}$, $b_2 \in \mathbb{R}^{10}$, and $T$ is the temperature for smoothing the output of first layer. For data generation, $T = 2$.

For layer $i \in \{1, 2\}$, we model $W_k^i \sim \mathcal{N}(u_k^i, 1)$, $b_k^i \sim \mathcal{N}(u_k^i, 1)$, $u_k^i \sim \mathcal{N}(0, \alpha)$, and $x_k \sim \mathcal{N}(v_k, \Sigma)$, where $\Sigma$ is a diagonal covariance matrix with $\Sigma_{(j,j)} = j^{-1.2}$. Each element in the mean vector $v_k$ is drawn from $\mathcal{N}(B_k, 1)$, with $B_k \sim \mathcal{N}(0, \beta)$. Therefore, $\alpha$ controls the variation among local models, while $\beta$ controls the differences between local data across clients. We vary $\alpha$, $\beta$ to generate two levels of heterogeneous datasets, with $(\alpha, \beta)$ set to $(0, 0)$ and $(0.5, 0.5)$.

# B  Hyperparameter Details

The hyperparameters studied in each of the experiments, along with their optimal values, are outlined in this section. For each dataset, the only hyperparameter tuned when using FedAvg is the learning rate (LR) for their respective local optimizers. The LR is selected from the set $\{0.00001, 0.0001, 0.001, 0.01, 0.1\}$. The optimal hyperparameters found for each method and dataset are displayed in Table 3. For any fixed set of parameters, three runs are performed with random seeds of $\{2021, 2022, 2023\}$.

## B.1  CIFAR-10

Following the implementation in pFL-bench [3], we use an SGD optimizer with momentum 0.9. For all levels of heterogeneity, the same parameter ranges are considered. For Ditto, the client-side LR are $\{0.0001, 0.001, 0.01, 0.1\}$ and $\lambda \in \{0.1, 0.5, 1, 2, 10\}$.

For cosine similarity, when applied alone, the LR values are $\{0.001, 0.01\}$, and $\mu$ has values $\{0.01, 0.1, 1.0\}$. When combined with Ditto, LR $\in \{0.001, 0.01\}$, $\mu \in \{0.01, 0.1, 1.0\}$, and $\lambda \in \{0.1, 0.5, 1, 2, 10\}$.

In experiments applying only MMD-D and MK-MMD, the LR is selected from $\{0.001, 0.01\}$. The value of $\mu$ is drawn from $\{0.01, 0.1, 1.0\}$. Constraint depths of 1, 2, and 3 are considered with the optimal depth always being 1. As such, when combining the standard Ditto penalty and an MMD-based measure, a depth of 1 is always used. In such settings, the LR values are $\{0.001, 0.01\}$, $\mu$ is drawn from $\{0.01, 0.1, 1.0\}$, and $\lambda$ is selected from $\{0.1, 0.5, 1, 2, 10\}$.

## B.2  Synthetic

Following the setup in [20], an SGD optimizer with momentum 0.9 and weight decay 0.001 is used. For both levels of heterogeneity, the same parameter ranges are considered. For Ditto, the LR is selected from $\{0.0001, 0.001, 0.01, 0.1\}$ and $\lambda \in \{0.01, 0.1, 1.0\}$.

In the remaining experiments, based on the Ditto results, the LR is fixed to 0.001 for $(\alpha, \beta) = (0, 0)$ and to 0.01 for data with $(\alpha, \beta) = (0.5, 0.5)$. The hyperparameter $\mu$ for cosine similarity, MMD-D, and MK-MMD is selected from the set $\{0.01, 0.1, 1.0\}$. When these methods are combined with Ditto, we additionally select $\lambda$ from $\{0.01, 0.1\}$ while continuing to tune $\mu$ within the same range.

## B.3  Fed-ISIC2019

In [39], using an AdamW optimizer for this dataset led to improved performance in the FL setup, compared to using SGD, as was done in [8]. Based on these findings, AdamW is used in the experiments. Similar to the previous two datasets, for Ditto we search for the optimal LR in the

Table 3: Best hyperparameters for each of the methods across all datasets.

| | | CIFAR-10 | | | Synthetic | | Fed-ISIC2019 | RxRx1 |
|---|---|---|---|---|---|---|---|---|
| | | 0.1 | 0.5 | 5.0 | $(0.0, 0.0)$ | $(0.5, 0.5)$ | – | – |
| FedAvg | LR | 0.01 | 0.01 | 0.01 | 0.001 | 0.01 | 0.0001 | 0.0001 |
| Ditto | LR | 0.01 | 0.01 | 0.01 | 0.001 | 0.01 | 0.001 | 0.0001 |
| | $\lambda$ | 0.1 | 0.1 | 0.5 | 0.01 | 0.1 | 1.0 | 0.01 |
| MMD-D | LR | 0.01 | 0.01 | 0.01 | 0.001 | 0.01 | 0.001 | 0.0001 |
| | $\mu$ | 0.01 | 0.01 | 0.01 | 1.0 | 1.0 | 1.0 | 0.01 |
| MK-MMD | LR | 0.01 | 0.01 | 0.01 | 0.001 | 0.01 | 0.001 | 0.0001 |
| | $\mu$ | 1.0 | 1.0 | 1.0 | 1.0 | 0.1 | 1.0 | 0.01 |
| Cos. Sim. | LR | 0.01 | 0.01 | 0.01 | 0.001 | 0.01 | 0.001 | 0.0001 |
| | $\mu$ | 0.1 | 0.1 | 0.1 | 1.0 | 1.0 | 1.0 | 1.0 |
| Ditto + MMD-D | LR | 0.01 | 0.01 | 0.01 | 0.001 | 0.01 | 0.001 | 0.0001 |
| | $\lambda$ | 0.1 | 0.1 | 0.5 | 0.01 | 0.01 | 1.0 | 0.01 |
| | $\mu$ | 0.01 | 0.01 | 0.01 | 1.0 | 1.0 | 1.0 | 0.01 |
| Ditto + MK-MMD | LR | 0.01 | 0.01 | 0.01 | 0.001 | 0.01 | 0.001 | 0.0001 |
| | $\lambda$ | 0.1 | 0.1 | 1.0 | 0.01 | 0.01 | 1.0 | 0.01 |
| | $\mu$ | 0.1 | 0.1 | 0.1 | 0.1 | 1.0 | 0.1 | 0.1 |
| Ditto + Cos. Sim. | LR | 0.01 | 0.01 | 0.01 | 0.001 | 0.01 | 0.001 | 0.0001 |
| | $\lambda$ | 0.1 | 0.1 | 0.5 | 0.01 | 0.01 | 0.1 | 0.01 |
| | $\mu$ | 0.1 | 0.1 | 1.0 | 1.0 | 1.0 | 0.1 | 0.01 |

set $\{0.0001, 0.001, 0.01, 0.1\}$, and tune $\lambda$ over $\{0.1, 1\}$. Among these, an LR of 0.001 consistently outperformed other values.

Fixing the LR to 0.001, the hyperparameter $\mu$ is tuned for cosine similarity, MMD-D, and MK-MMD from the set $\{0.01, 0.1, 1.0\}$. When these methods are integrated with Ditto, the same range for $\mu$ is searched, along with $\lambda \in \{0.1, 1\}$.

### B.4   RxRx1

Following [4], we use an AdamW optimizer to fine-tune a pretrained ResNet-18 model. For Ditto, the LR is tuned within the set $\{0.00001, 0.0001, 0.001, 0.01\}$, and the regularization parameter $\lambda$ is taken from $\{0.01, 0.1, 1\}$. An LR of 0.0001 consistently yielded the best performance across various $\lambda$ values.

With the learning rate fixed at 0.0001, we optimize the hyperparameter $\mu$ for cosine similarity, MMD-D, and MK-MMD from the set $\{0.01, 0.1, 1.0\}$. When these methods are combined with Ditto, $\lambda = 0.01$ and the same range of search for an optimal $\mu$ is maintained.

## C   FL Settings and Checkpointing

In this section, dataset specific FL settings for the experiments are outlined and the checkpointing strategy is described. Regardless of dataset, each client participates in every round of FL training. That is, there is no client subsampling. It should be noted that, in principle, the Ditto approach allows for different types of optimizers, and learning rates, to be applied to the global and local models during client-side optimization. However, in the experiments, the same LR and optimizer type are applied to both models, which is also done in the original work.

For all MK-MMD experiments, in constructing the kernel space $\mathcal{K}$, $d = 18$ different RBF kernels are used of the form $k_j(x, y) = e^{-\gamma_j \|x-y\|_2^2}$. The values of $\{\gamma_j\}_{j=1}^d$ ranged from $2^{-3.5}$ to $2^1$ with exponents evenly spaced in increments of 0.25. Further, when periodically optimizing MK-MMD or MMD-D every 20 steps, the number of batches of data used for optimization, $z$, varied. For Fed-ISIC2019, $z = 64$, and for the remainder of datasets $z = 50$.

For all variants of CIFAR-10, there are five clients and FL proceeds for 10 server rounds. Within each round, clients perform five epochs of local training using a standard SGD optimizer with momentum set to 0.9 and a batch size of 32. For the Synthetic data experiments, we generate eight clients, each with 5000 samples. Federated training is conducted over 15 server rounds, with each round consisting of 5 local training epochs. Following [20], we use a batch size of 10 and optimize the models using SGD with a momentum of 0.9 and a weight decay of 0.001. Following the settings of the FLamby benchmark [8], there are 15 rounds of federated training with six participating clients for Fed-ISIC2019. During an FL round, each client trains for 100 steps using a batch size of 64. An AdamW optimizer is used with default parameters. Finally, for RxRx1, four clients are present and 10 rounds of federated training are run. Each client performs five epochs of local training using a batch size of 32. Again an AdamW optimizer is applied using default parameters.

In all experiments, the final model for each client is checkpointed and evaluated on a held-out test set to quantify performance. This is a commonly applied approach in FL [20, 14, 31]. For FedAvg, the final global (aggregated) model is saved. For all other approaches, the local models are persisted. During experimentation, we also considered checkpointing each client's model after each round of local training based on local validation performance. Because results were similar or worse for all approaches with such checkpointing, only those using the latest checkpoint are presented.

## D   Compute Resources and Runtime Complexity: Ditto Experimentation

All experiments were performed on a high-performance computing cluster. For the CIFAR-10 and Synthetic datasets, an NVIDIA T4V2 GPU with 32GB of CPU memory was used. Each FedAvg and Ditto experiment on CIFAR-10 took approximately 10 and 20 minutes, respectively. The addition of optimization-based losses increased training time, with MMD-D, MK-MMD$_{-1}$, and MK-MMD$_{20}$ experiments requiring approximately 2, 1, and 1.5 hours, respectively. Training for the Synthetic data took under 5 minutes on the same GPU for all methods.

For the Fed-ISIC2019 and RxRx1 datasets, we used an NVIDIA A100 GPU with 64GB and 100GB of CPU memory, respectively. With Fed-ISIC2019, FedAvg and Ditto required 2 and 3.5 hours of training time, while incorporating MMD-D, MK-MMD$_{-1}$, and MK-MMD$_{20}$ increased the training time to approximately 4, 4, and 7 hours, respectively. For RxRx1, FedAvg took 2 hours, Ditto took 2.5 hours, and the addition of MMD-D, MK-MMD$_{-1}$, and MK-MMD$_{20}$ extended training times to 3.5, 3, and 6 hours, respectively.

The objective of this work is demonstrating that integrating the proposed MMD measures improves performance in heterogeneous data settings. Thus, no concerted effort is made to reduce the overhead associated with optimization of the MK-MMD or MMD-D kernels. However, a number of possibilities for cost reduction exist. For MK-MMD, a general, python-based, quadratic program (QP) solver called `qpth` is used. Experimenting with more heavily optimized libraries or QP solvers specifically tailored to the MK-MMD system may yield large reductions. It is also possible that the QP systems need not be solved to significant precision, decreasing the number of iterations used by the solvers. Finally, pre-conditioning techniques, such as using previous solutions as iteration starting points could markedly compress computation time. For MMD-D, future work could include studying utility trade-offs associated with various configurations, including kernel update frequencies, training data sizes, and optimizers. In the experiments, the featurization networks, $\varphi$, are also of moderate size, consisting of four linear layers and activations with a hidden dimension of 50. Contracting or changing this architecture would likely speed up optimization.

### D.1   Runtime Complexity Analysis

The use of the proposed, adaptive MMD measures comes with an increase in runtime complexity due to the arithmetic operations associated with either solving the quadratic program for MK-MMD or performing SGD steps with MMD-D. In this section, we quantify these increases using Ditto as the reference algorithm. Borrowing the notation of Algorithm 1, let $p$ denote the number of arithmetic operations for a single step of SGD with batch size $b$ to update the model weights, $w_L$ or $w_G$. For Ditto, the total cost across all $N$ clients is $\mathcal{O}(N \cdot T \cdot s \cdot p)$. For simplicity, assume we perform kernel optimization for either MK-MMD or MMD-D after every iteration of local model training and have a single latent-space penalty.

For MMD-D, let $q$ be the arithmetic operations associated with a single step of SGD for the deep kernel, $\varphi$, with batch size $b$. Denote by $k$ the number of kernel optimization steps taken. Then the total cost across all clients becomes $\mathcal{O}(N \cdot T \cdot s \cdot (p + kq))$. In most cases, $p$ will be significantly greater than $kq$, as the primary models are expected to be much larger than $\varphi$.

With MK-MMD, a convex quadratic program needs to be constructed according to Equation (2). However, the cost of this construction is generally smaller than the cost of solving the system for small data sizes. A QP solver based on interior-point methods requires $\mathcal{O}(\sqrt{m})$ iterations, each costing $\mathcal{O}(m^3)$ arithmetic operations, where $m$ is the size of the latent space [43]. Hence, the total cost across all clients is $\mathcal{O}(N \cdot T \cdot s \cdot (p + m^{3.5}))$. As in the case of MMD-D, the quantity $m^{3.5}$ will often be much smaller than $p$, as $d$ simply represents the dimension of the constrained latent space.

## E   Extended Related Work: Other pFL Approaches

This section discusses less similar, but still related, approaches in pFL and how the measures proposed in this work might be incorporated, if applicable. Sequentially split models are common in pFL. Quintessential examples include FedPer and FedRep [1, 5]. These techniques sequentially decompose models into feature maps and classifier layers and only exchange model sub-components, most often the feature maps, with the server for aggregation. As a result, only certain modules incorporate global information through aggregation. Parallel-split pFL methods employ a similar trick, but decompose the models differently. In these methods, two or more modules process input in parallel. At least one of the modules is meant to be aggregated, incorporating global information, while others remain strictly local. Parallel modules could constitute complete classifiers, as in APFL [6], or lower-level feature maps, as in PerFCL [46] or FENDA-FL [39]. Combining partial model exchange, whether sequential or parallel, with penalty-based pFL approaches like Ditto can take many forms, including straightforward modifications. For example, during local training, the feature maps in FedRep may be constrained to not drift too far from those of a FedAvg model, as in Ditto, using the MK-MMD or MMD-D measures proposed in this work.

Other pFL methods, fitting into less broad categories, exist. FedBN [22] proposes excluding batch normalization layers from global exchange, as these layers can struggle to handle heterogeneity well. HypCluster [27] partitions clients into clusters and trains isolated models for each cluster. When the number of clients is small, as in this work, such partitioning is impractical and forgoes global information. Other methods, such as FedPop [15], are explicitly designed for cross-device settings rather than cross-silo FL. They take advantage of large-scale client populations with relatively small datasets, rather than focusing on smaller client pools. Each of these techniques incorporate ideas that narrow their applicability with respect to the setting considered in this work.

## F   MR-MTL Experiments

The integration of the proposed adaptive MMD measures is not limited to the Ditto algorithm. This appendix demonstrates the extensibility of the main results to other existing approaches. In the experiments below, MR-MTL is augmented to include either MK-MMD or MMD-D as latent-space penalties. The training process of MR-MTL is similar to Ditto except that, instead of training a separate global model to constrain local model training, the local models are simply penalized when drifting too far from the average of all local models [24]. Thus, the algorithm resembles FedProx, but without replacing local models with the averaged one prior to client-side training.

While MR-MTL is competitive with Ditto for certain tasks, it tends to be a slightly less effective approach. However, the removal of the separate global-model training process has significant differential-privacy benefits and also markedly reduces the computational cost that comes with training two models simultaneously [24]. Thus, MR-MTL is useful in settings where privacy is of the utmost importance or compute constraints intervene.

Table 4 summarizes the experimental results. As observed with Ditto, combining the weight-based $\ell^2$ constraint of MR-MTL with the adaptive, latent-space constraints proposed in this work has visible benefits. For all variants of CIFAR-10 and the RxRx1 dataset, using either MK-MMD or MMD-D markedly boosts performance. Similarly, the Synthetic datasets also see improvements with MMD-D for both heterogeneity levels, and MK-MMD lifts performance when $\alpha = \beta = 0.5$. Fed-ISIC2019 is the lone dataset that does not see improvements when augmenting with these constraints.

Table 4: Average performance (and standard deviation) when **augmenting** the MR-MTL constraint with the proposed MMD penalties. Bold and underline indicate the best and second best value. Subscripts for CIFAR-10 and Synthetic indicate values for $\alpha$ and $\alpha = \beta$, respectively.

| Dataset | MR-MTL | + MK-MMD$_{20}$ | + MMD-D |
|---|---|---|---|
| CIFAR-10$_{0.1}$ | 79.516 (1.674) | **81.269** (0.400) | 80.307 (1.705) |
| CIFAR-10$_{0.5}$ | 73.361 (1.027) | 74.333 (0.742) | **74.446** (0.529) |
| CIFAR-10$_{5.0}$ | 68.224 (1.223) | 69.487 (1.180) | **70.353** (0.149) |
| Synthetic$_{0.0}$ | 90.879 (0.267) | 90.708 (0.257) | **91.142** (0.031) |
| Synthetic$_{0.5}$ | 86.750 (3.189) | **90.958** (0.981) | 90.337 (0.499) |
| Fed-ISIC2019 | **70.628** (0.911) | 68.180 (2.735) | 70.366 (0.970) |
| RxRx1 | 64.065 (1.663) | 65.791 (0.363) | **66.673** (0.480) |

The setup for the MR-MTL experiments mirrors that of Ditto. The same FL settings and checkpointing described in Appendix C are used, and the same models are trained. The values reported in Table 4 are the average of three distinct runs. The optimal hyperparameters for each method and dataset are displayed in Table 5. The hyperparameter ranges swept are reported thereafter. Note that the ranges are subsets of those covered in the Ditto experiments, as only neighborhoods of the optimal values there are considered.

Table 5: Best hyperparameters for methods in the MR-MTL experiments across all datasets.

| | | CIFAR-10 | | | Synthetic | | Fed-ISIC2019 | RxRx1 |
|---|---|---|---|---|---|---|---|---|
| | | 0.1 | 0.5 | 5.0 | (0.0, 0.0) | (0.5, 0.5) | – | – |
| MR-MTL | LR | 0.001 | 0.001 | 0.001 | 0.001 | 0.001 | 0.0001 | 0.0001 |
| | $\lambda$ | 0.1 | 0.1 | 0.1 | 0.01 | 0.01 | 0.1 | 0.01 |
| + MMD-D | LR | 0.001 | 0.001 | 0.001 | 0.001 | 0.001 | 0.0001 | 0.0001 |
| | $\lambda$ | 0.1 | 0.1 | 0.1 | 0.01 | 0.01 | 0.1 | 0.01 |
| | $\mu$ | 1.0 | 1.0 | 1.0 | 1.0 | 1.0 | 0.01 | 0.01 |
| + MK-MMD$_{20}$ | LR | 0.001 | 0.001 | 0.001 | 0.001 | 0.01 | 0.0001 | 0.0001 |
| | $\lambda$ | 0.1 | 0.1 | 0.1 | 0.01 | 0.01 | 0.1 | 0.01 |
| | $\mu$ | 0.01 | 0.01 | 0.01 | 1.0 | 1.0 | 0.01 | 0.1 |

**CIFAR-10** For all levels of heterogeneity, the same parameter ranges are used. For MR-MTL, the client-side LRs are $\{0.001, 0.01, 0.1\}$ and $\lambda \in \{0.1, 0.5, 1.0\}$. In experiments applying either MMD-D and MK-MMD alongside the traditional MR-MTL constraint, the LR is fixed at 0.001, $\lambda = 0.1$, and $\mu$ is drawn from $\{0.01, 0.1, 1.0\}$.

**Synthetic** For both levels of heterogeneity, the same parameter ranges are searched. For MR-MTL, the LR is $\{0.001, 0.01\}$ and $\lambda \in \{0.01, 0.1, 1.0\}$. When MK-MMD or MMD-D are combined with MR-MTL, we fix $\lambda = 0.01$ while tuning $\mu \in \{0.1, 1.0\}$ and LR in $\{0.001, 0.01\}$.

**Fed-ISIC2019** For MR-MTL, we search for the optimal LR in the set $\{0.0001, 0.001\}$, and tune $\lambda$ over $\{0.01, 0.1, 1\}$. When MK-MMD or MMD-D are used with MR-MTL, the LR is constant at 0.0001 and $\lambda = 0.1$. The value of $\mu$ is tuned in the range $\{0.01, 0.1, 1.0\}$.

**RxRx1** For MR-MTL, the LR is tuned within the set $\{0.0001, 0.001\}$, and $\lambda$ is taken from $\{0.01, 0.1, 1.0\}$. For a fixed learning rate of 0.0001 and $\lambda = 0.01$, the hyperparameter $\mu$ is optimized, when combining MR-MTL and MMD-D or MK-MMD, within $\{0.01, 0.1, 1.0\}$.

## G   Standard Deviations of Main Results

Due to space constraints, and for clarity of presentation, the standard deviations of the main results of Section 6 are deferred to this appendix. Table 6 reports a condensed version of Tables 1 and 2

Table 6: Average performance (and standard deviations) for various constraint configurations. Bold indicates the best values. Subscripts for CIFAR-10 and Synthetic indicate values for $\alpha$ and $\alpha = \beta$, respectively. Finally + indicates that the MMD constraint is applied in tandem with the standard Ditto constraint, rather than as a replacement.

| Dataset | Ditto | MMD-D | MK-MMD$_{-1}$ | MK-MMD$_{20}$ |
|---|---|---|---|---|
| CIFAR-10$_{0.1}$ | **84.930** (0.142) | 83.789 (0.184) | 84.136 (0.346) | 84.439 (0.304) |
| CIFAR-10$_{0.5}$ | **80.702** (0.121) | 75.094 (0.532) | 75.678 (0.000) | 76.564 (0.125) |
| CIFAR-10$_{5.0}$ | **77.658** (0.268) | 67.729 (0.591) | 68.718 (0.262) | 68.832 (0.174) |
| Synthetic$_{0.0}$ | 89.129 (2.200) | 90.237 (0.405) | **91.418** (0.339) | 90.066 (2.628) |
| Synthetic$_{0.5}$ | 85.533 (0.250) | **91.270** (0.309) | 91.137 (0.694) | 90.262 (0.144) |
| Fed-ISIC2019 | **71.350** (1.191) | 64.302 (1.546) | 60.168 (2.979) | 62.677 (0.604) |
| RxRx1 | 65.629 (1.936) | **67.478** (1.200) | 65.861 (1.151) | 67.078 (0.394) |
| Dataset | Ditto | + MMD-D | + MK-MMD$_{-1}$ | + MK-MMD$_{20}$ |
| CIFAR-10$_{0.1}$ | 84.930 (0.142) | **85.214** (0.079) | 84.723 (0.501) | 84.900 (0.318) |
| CIFAR-10$_{0.5}$ | 80.702 (0.121) | 80.696 (0.148) | 80.936 (0.224) | **80.976** (0.365) |
| CIFAR-10$_{5.0}$ | 77.658 (0.268) | **77.739** (0.376) | 77.578 (0.139) | **77.739** (0.271) |
| Synthetic$_{0.0}$ | 89.129 (2.200) | **89.458** (0.348) | 89.183 (2.249) | 89.258 (2.269) |
| Synthetic$_{0.5}$ | 85.533 (0.250) | **89.695** (0.186) | 88.154 (1.817) | 88.104 (0.721) |
| Fed-ISIC2019 | 71.350 (1.191) | **72.226** (1.241) | 71.169 (1.369) | 71.267 (1.037) |
| RxRx1 | 65.629 (1.936) | **67.755** (0.672) | 65.984 (0.774) | 66.892 (0.851) |

with the corresponding standard deviations over three runs. It should be noted that in pFL, especially under heterogeneous settings, a certain degree of performance variation is expected and has also been reported in previous work [29].

# H  Initial Thoughts on Optimal Conditions for Adaptive MMD Measures

This work empirically demonstrates that the proposed adaptive, feature-based MMD penalties are beneficial in the context of pFL algorithms imposing local drift constraints. The experiments are specifically designed to provide insight into settings where such penalties provide the most benefit. The results suggest that the presence of strong feature heterogeneity is one such setting. In contrast, when label heterogeneity is induced via Dirichlet allocation and training data is otherwise independent and identically distributed (IID), as in the CIFAR-10 datasets, the benefits are less pronounced.

In most settings, theoretical support for FL algorithms centers on demonstrating convergence through bounding the magnitude of model updates or local gradients by some contracting value with respect to the server rounds [20, 38, 21, 32]. Proving various aspects of optimality compared with other algorithms is more rare, especially in non-IID settings. Some exceptions of interest are the convergence rates analysis of SCAFFOLD [14] and the optimal regularization parameter theory of Ditto and MR-MTL [21, 24].

In [21], it is shown that there exists a regularization parameter value, $\lambda^*$, yielding optimal average test performance across heterogeneous clients for the Ditto framework in the context of federated linear regression. Beyond the simplified model setting, the setup assumes that clients' heterogeneity is expressed as perturbations in their labeling functions from a common mapping. While this heterogeneity is different in kind from the feature heterogeneity for which the approach proposed herein empirically improves performance, the theoretical model may be useful in building similar results for latent-space penalties.

A possible avenue is considering a shared latent-space distribution and labeling function across clients but varying linear feature maps. As such, the input feature spaces of each client would have differing statistical properties induced by the inverse feature maps. Nonetheless, with the correct feature mappings, labels are consistently and accurately applied. Provided that the target latent-space

has structured statistical properties, such as being a Gaussian mixture, and the feature maps are sufficiently conditioned or simple, deriving optimal MMD kernels may be possible. From there, showing that there exists an optimal weight, $\mu$, for such a kernel would yield the desired result.

The setting above is specifically manufactured to be difficult for a standard Ditto-based approach. By construction, feature maps are not within a neighborhood of one another, implying that penalizing drift from a central model is likely detrimental. While the setting is artificial, it is related to real problems, where, for example, different medical imaging devices produce unique artifacts but fundamentally similar decision criterion when these artifacts are accounted for during feature processing.

Characterizing the settings under which the MMD constraints studied in this work are optimal remains an open and interesting question. While the suggested theoretical tack may require modifications, the experimental results in this paper provide insights into the design of such theoretical machinery.

