# OpenReview forum: "Adaptive Latent-Space Constraints in Personalized Federated Learning"
_NeurIPS.cc/2025/Conference — NeurIPS 2025 poster_

### Official Review · Reviewer_Vrrq · 2025-06-13

**Clarity:** 3
**Significance:** 3
**Originality:** 3
**Rating:** 4
**Confidence:** 3

**Summary:**

This paper proposes an improvement to the Ditto framework for personalized federated learning (pFL) by incorporating adaptive latent space constraints using statistical distance measures — specifically MK-MMD and MMD-D — instead of the traditional ℓ2 norm on weights. These constraints aim to better handle feature heterogeneity by minimizing distributional drift in the latent representation space rather than simple weight differences. The authors conduct experiments on diverse datasets (CIFAR-10, Synthetic, Fed-ISIC2019, RxRx1) and show that the proposed methods offer consistent improvements, particularly under significant heterogeneity. The approach is theoretically grounded and broadly applicable to other pFL frameworks beyond Ditto.

**Questions:**

1. More Experiments: Although the paper claims that the method has wide applicability, the current validation is limited to the Ditto framework. Experimental validation on other mainstream or different types of pFL frameworks would be more convincing.
2. Privacy Considerations: The reliance on feature embeddings for regularization raises potential privacy concerns. Specifically, intermediate activations can retain semantic information about raw inputs, which—if combined with model inversion attacks or a compromised global model—may lead to input reconstruction or data leakage.
3. Complexity Concern: How do the local computation cost and storage usage compare with Ditto and FedAvg? Is there a trade-off here?
4. Novelty Concern: As a mature analysis method, MMD is very likely to improve PFL. Although the authors have made some improvements, I am still concerned about the novelty of the insight and technical contribution. I hope the authors can explain their novelty more clearly.

I will consider increasing my score if my questions are answered.

**Ethical Concerns:**

["NO or VERY MINOR ethics concerns only"]

**Final Justification:**

Questions regarding the validation of the experimental framework and the novelty of the method have been addressed. The authors also reported a relatively honest comparison of their methods in terms of storage, computational overhead, and computational time. While the proposed method still has room for improvement in computational time, this could be a promising direction for future research. Overall, the proposed method is interesting, and the evaluation criteria used can effectively support innovation across different approaches. I improved my score after discussing this with the authors.

**Limitations:**

See Questions above.

**Paper Formatting Concerns:**

No.

**Quality:**

3

**Strengths And Weaknesses:**

Strengths：
 1. Clear Problem Focus and Innovation: The paper precisely identifies feature heterogeneity as a crucial challenge in Personalized Federated Learning (pFL). It proposes to address this by introducing adaptive, theoretically supported MMD constraints within an existing advanced framework (Ditto). This approach, building upon a strong baseline, is more convincing.
2. Constraints in Latent Space: Compared to imposing constraints in the raw data space, applying constraints in the latent space is a more advanced and flexible way to handle feature heterogeneity. It operates on abstract representations of data, promising to capture deeper distributional relationships.
3. Modularity and Extensibility: Although the research focuses on the Ditto framework, the paper suggests that these adaptive MMD constraints can be directly applied to other pFL settings. This indicates a modular design and good generality, facilitating broader adoption.

Weaknesses:
See Questions below.

---

> ### Author Rebuttal · Authors · 2025-07-31
>
> Thank you for your thoughtful review and providing us the opportunity to improve our work. Below is an itemized list of your questions, followed by a discussion and description of how they were addressed in revising the paper. We look forward to continuing to answer your questions about our work and further improving the presentation of our ideas.
>
> **More Experiments: Although the paper claims that the method has wide applicability, the current validation is limited to the Ditto framework. Experimental validation on other mainstream or different types of pFL frameworks would be more convincing.**
>
> We appreciate this feedback. To better support the conclusion that our proposed method is generalizable to other pFL algorithms, we have expanded our experimental results. Specifically, we augmented the MR-MTL algorithm to include the MK-MMD and MMD-D constraints and evaluated such integration across all seven datasets. The table below summarizes the results.
>
> | Dataset        | MR-MTL | MR-MTL+MK-MMD | MR-MTL+MMD-D |
> |----------------|--------|---------------|------------------|
> | **CIFAR (0.1)**      | 79.515 | **81.268**        | 80.307           |
> | **CIFAR (0.5)**     | 73.360 | 74.332        | **74.446**           |
> | **CIFAR (5.0)**      | 68.224 | 69.487        | **70.352**           |
> | **Synth (0.0)**  | 90.879 | 90.708        | **91.141**          |
> | **Synth (0.5)**  | 86.750 | **90.958**        | 90.337           |
> | **Fed-ISIC2019**    | **70.627** | 68.180        | 70.366           |
> | **RxRx1**          | 64.064 | 65.790        | **66.673**           |
>
> Both MK-MMD and MMD-D variants largely improve performance compared to the baseline, MR-MTL, with the exception of the Fed-ISIC2019 dataset. We believe these results better support our claim of transferability and also provide more evidence to the effectiveness of our approach in general.
>
> **Privacy Considerations: The reliance on feature embeddings for regularization raises potential privacy concerns. Specifically, intermediate activations can retain semantic information about raw inputs, which—if combined with model inversion attacks or a compromised global model—may lead to input reconstruction or data leakage.**
>
> Privacy is an extremely important consideration in FL, and there are downsides to the Ditto approach in terms of privacy guarantees due to the additional data iterations associated with simultaneously training two models, as noted in [1]. However, the method proposed here does not modify the information exchanged between the clients and server nor does it expose the feature embeddings of individual clients to others. As such, the privacy implications are nearly identical to that of Ditto, with the exception that training data is periodically used to optimize the MMD kernels. We have noted this in Section 3.1 as an important point.
>
> **Complexity Concern: How do the local computation cost and storage usage compare with Ditto and FedAvg? Is there a trade-off here?**
>
> This is a good question. Throughout the manuscript, we have endeavored to make the communication,  computation, and storage costs associated with our work more clear.  In our original submission, we did discuss computational overhead compared to FedAvg and Ditto in Appendix D, but we failed to refer to that section in the main body. So it was not well-represented to readers. We now include a reference to these details in Section 7. We have also expanded the Appendix to include a more thorough discussion of ways to reduce computational cost, as the present overhead is fairly high.
>
> The communication costs associated with the approach are identical to Ditto, as no changes are made to the information exchanged. We now explicitly note this in Section 3.1. The memory overhead of the proposed method is also nearly identical to Ditto, with the exception of storing a modest number of feature activations and training a set of kernel weights. We have added a remark on this to Appendix D as well.
>
> **Novelty Concern: As a mature analysis method, MMD is very likely to improve PFL. Although the authors have made some improvements, I am still concerned about the novelty of the insight and technical contribution. I hope the authors can explain their novelty more clearly.**
>
> Thank you for the opportunity to better describe our contributions. The technical contributions in this paper can be summarized as follows.
>
> 1. We propose the first use of adaptive and theoretically supported measures as penalty constraints for dealing with heterogeneity in pFL, moving beyond previously considered paired feature-based constraints, using Ditto, and now MR-MTL, as representative examples.
>
> 2. While a small collection of existing work applies MMD in different ways outside of pFL, all of them used fixed kernels. We demonstrate that taking advantage of the adaptability of MK-MMD and MMD-D measures through periodic re-optimization during training provides notable performance improvements in settings with high feature heterogeneity, where existing methods, such as Ditto, under-perform.
>
> 3. Finally, experiments with either natural or controllable levels of label and feature heterogeneity highlight the strengths and weaknesses of the types of drift penalties investigated.
>
> We have endeavored to make these novel contributions more clear throughout the manuscript and hope that the presentation has been significantly improved.
>
> [1] Ziyu Liu, Shengyuan Hu, Zhiwei Steven Wu, and Virginia Smith. On privacy and personalization in cross-silo federated learning.

---

> > ### Comment · Reviewer_Vrrq · 2025-08-07
> > **Reply.**
> >
> > Thank you for the response.
> >
> > Questions regarding the validation of the experimental framework and the novelty of the method have been addressed.
> >
> > I am inclined to accept the paper.
> >
> > I also note that the computational time of this method is longer than that of the baseline method. I hope to see a detailed and clear discussion of the computational effort, computational time, and storage in the updated main body of the article.

---

> > > ### Author Response · Authors · 2025-08-07
> > >
> > > It is great to hear that our response and new experiments have addressed your questions. Should this work move forward, a thorough discussion of the various aspects of overhead incurred by the proposed algorithm will be integrated into the main body of the manuscript.
> > >
> > > Thank you again for your engagement with our work. Your feedback has truly helped us improve the paper.

---

### Official Review · Reviewer_AkTQ · 2025-06-29

**Clarity:** 3
**Significance:** 2
**Originality:** 1
**Rating:** 4
**Confidence:** 3

**Summary:**

This paper introduces adaptive statistical constraints - specially Multi Kernel Maximum-mean Discrepeancies (MK-MMD) and deep MMD (MMD-D) - to the panelty-based (e.g., Ditto) personalized federated learning (PFL) framework. It empirically demonstrates that the proposed methods improve performance in tasks with feature and label heterogeneity compared to existing feature-space and weight-based constraints.

However, this paper’s contribution seems incremental. It essentially applies well-established statistical distance measures (MK-MMD, MMD-D) to the latent-space distributions within Ditto, without clear theoretical innovation. The justification to choose MMD is not established, and empirical results shows that performance gain from using MMD is not significant.

**Questions:**

- 1. What is the difference of the proposed methods from the prototype based PFL? As the prototype-based PFL also apply constraints in the latent space, discussion about this should be included in the related works.
- 2. Why should MMD be combined to the penalty-based pFL (e.g. Ditto). It seems that applying MMD is orthogonal to the penalty-based constraints.
- 3.  In section 3.1, the authors claim that this paper proposes an unpaired approach. In MMD-D, however, feature-drift constraints depicted in figure 1 seems to be paired. Please correct me if I missed something.
- 4. What is commputational and communication costs of the proposed schemes to additionally apply MMD on top of Ditto?

**Ethical Concerns:**

["NO or VERY MINOR ethics concerns only"]

**Final Justification:**

My previous concerns have been addressed by the author's detailed explanation and additional experiments, and hence I increased my score. I still believe that the paper would be strengthened by including an intuitive discussion of why combining latent-space constraints with weight-based constraints creates synergy, especially in feature-heterogeneous FL.

**Limitations:**

yes

**Paper Formatting Concerns:**

There is no formatting issues.

**Quality:**

2

**Strengths And Weaknesses:**

- Strenghts
  - 1. This paper considers important problems in PFL, feature-drift.
  - 2. This paper is well written and hence is easy to follow.

- Weaknesses
  - 1. The major concern of the reviewer is the novelty of the paper. MMD constraints are not new, nor is their application in deep learning contexts unusual. Merely applying these known measures to Ditto feels straightforward rather than innovative. The authors must articulate precisely what novel insight or substantial methodological development distinguishes this work from straightforward integration.
  - 2. The authors rely heavily on empirical validation without providing theoretical conditions for why MMD constraints are the best or good to be combined with the penalty-based PFL such as Ditto.
  - 3. Even in the emprifical validation, performance gain from using MMD is not significant. Especially, in table 2, comparing to Ditto and the proposed schemes, the gain is less than 1% in most cases (only synthetic_0.5 has the large gain).
  - 4. While the proposed schemes has extra communications and computation (to exchange  $f_i(x;\bar{\theta_i})$ for all $i$ compared to the baseline (e.g., Ditto), this paper lacks computational and communication efficiency analysis, which weakens claims of practical utility.

---

> ### Author Rebuttal · Authors · 2025-07-31
>
> We would like to thank you for your time and effort with our manuscript. Your feedback thus far has motivated a number of improvements to our work. Below is an itemized list of your questions and remarks, followed by a discussion and description of how they were addressed in revising the paper. We look forward to further discussing our work with you and any additional questions you may have.
>
> **The major concern of the reviewer is the novelty of the paper. MMD constraints are not new, nor is their application in deep learning contexts unusual. Merely applying these known measures to Ditto feels straightforward rather than innovative. The authors must articulate precisely what novel insight or substantial methodological development distinguishes this work from straightforward integration.**
>
> We appreciate the chance to make the novelty of our contributions more clear.
>
> While MMD measures are well-established in other contexts, they are an under-explored tool in pFL. In this work, we propose the first use of adaptive MMD measures as latent-space penalties for dealing with heterogeneity in pFL, moving beyond previously considered paired feature-based penalties. Ditto is investigated as a representative example, but the use of such measures readily generalizes to other pFL settings.
>
> A small collection of existing work incorporates MMD into FL algorithms, outside of pFL, but all of them use fixed kernels. We demonstrate that taking advantage of the inherent adaptability of MK-MMD and MMD-D measures through periodic re-optimization during training provides notable performance improvements in settings with high feature heterogeneity, where existing methods under-perform.
>
> Finally, experiments with either natural or controllable levels of label and feature heterogeneity highlight the strengths and weaknesses of the types of drift penalties investigated, providing a starting point for understanding when the proposed measures perform most effectively.
>
> In the edits to our paper, we have striven to state these contributions clearly and to distinguish our work from the existing literature. We hope the above discussion provides a concise and direct summary of this work’s novelty.
>
> **The authors rely heavily on empirical validation without providing theoretical conditions for why MMD constraints are the best or good to be combined with the penalty-based PFL such as Ditto.**
>
> This is indeed a limitation that we recognize in the conclusion of our work. There is some existing theory around the weight-based penalty of Ditto that may serve as a good starting point. However, a complete characterization of settings where MMD-based latent-space penalties become optimal is, unfortunately, beyond the scope of this work. However, we have endeavored to construct our experiments in such a way as to suggest settings where our proposed latent-space penalties are most useful. Those being settings where datasets are anticipated to incorporate feature heterogeneity, as in the Synthetic, RxRx1, and Fed-ISIC2019 datasets.
>
> **Even in the empirical validation, performance gain from using MMD is not significant. Especially, in table 2, compared to Ditto and the proposed schemes, the gain is less than 1% in most cases (only synthetic_0.5 has the large gain).**
>
> The gains with the MMD measures are modest for the CIFAR-10 dataset, which considers synthetically induced label heterogeneity rather than feature heterogeneity. The accuracy gains for both Synthetic datasets and RxRx1 exceed 2%. The improvement for Fed-ISIC2019 is almost 0.9%, which is somewhat smaller, but still a notable improvement for a clinical task. In many benchmarks, the margins between the top performing methods are often less than 1%, and it is rare for an approach to uniformly improve performance, which we see here [1,2].
>
> The table below summarizes the improvements over Ditto with the best performing MMD approach across the datasets. We hope that this discussion provides more confidence in the significance of our observed improvements.
>
> | Dataset        | CIFAR (5.0) | CIFAR (0.5) | CIFAR (0.1) | Synth (0.0) | Synth (0.5) | Fed-ISIC2019  | RxRx1 |
> |----------------|-------------|-------------|-------------|--------------|--------------|--------|--------------|
> | **Acc. Gain**      | +0.283      | +0.274      | +0.081      | +2.289       | +5.885       |  +0.876  |   +2.126   |
>
> **While the proposed schemes has extra communications and computation (to exchange
>  f_i(x;\hat{\theta}_i) for all i) compared to the baseline (e.g., Ditto), this paper lacks computational and communication efficiency analysis, which weakens claims of practical utility.**
>
> A nice benefit of our proposed modification to Ditto is that no additional communication costs are introduced. As in the original Ditto formulation, only the global models are exchanged with the server for aggregation after client-side training. Each client maintains its own MMD kernel and optimizes it on local data. We realize that this could have been made more clear in the paper and have included remarks on communication and memory implications in Section 3.1.
>
> In our original submission, computational costs compared to FedAvg and Ditto were discussed in Appendix D. However, we failed to refer to that section in the main body, which reduced its visibility. A reference to these details now appears in Section 7. We have also expanded the appendix to include a more thorough discussion of potential ways to reduce computational cost.
>
> **What is the difference of the proposed methods from the prototype based PFL? As the prototype-based PFL also apply constraints in the latent space, discussion about this should be included in the related works.**
>
> We have refactored and expanded the Introduction and Related Works section to capture a wider swathe of pFL methods and better position our work within the existing literature. A primary addition to this is a discussion of prototype-based pFL methods, of which FedProto [3] is a canonical example. Prototype-based pFL methods aim to learn uniform feature representations for each data label. In FedProto, for example, only the prototypes are aggregated between clients, rather than model weights. In the approach here, we consider the distribution of all points in the feature space together. Prototype methods are also generally applicable to classification-based tasks only.
>
> Some prototype methods for pFL also have some notion of constraint. In FedProto, the local prototypes are constrained, via the $l^2$-norm, to not diverge too far from the aggregated prototypes. The method proposed here could be used to replace this norm as well.
>
> **Why should MMD be combined to the penalty-based pFL (e.g. Ditto). It seems that applying MMD is orthogonal to the penalty-based constraints.**
>
> This is a great question. We agree with the characterization of the two constraints as “orthogonal,” in some sense. It is perhaps why combining them can have a positive impact. In many natural datasets, it is unlikely that a single form of heterogeneity characterizes all drift between clients in an FL system. As such, we believe that incorporating losses that penalize drift in distinct ways can be beneficial and may be the source of improvements for the Fed-ISIC2019 and CIFAR-10 datasets when combining the two constraints. We have added additional discussion on this to the end of the Results section.
>
> **In section 3.1, the authors claim that this paper proposes an unpaired approach. In MMD-D, however, feature-drift constraints depicted in figure 1 seems to be paired. Please correct me if I missed something.**
>
> Thank you for pointing this out. We were ambiguous in our notation of $(\mathbf{x}, \mathbf{y})$ in Figure 1. This is meant to symbolize a batch of labeled data rather than a single input-label pair. We have updated the figure caption and the reference to figure in the text to make this clear.
>
> **What is computational and communication costs of the proposed schemes to additionally apply MMD on top of Ditto?**
>
> Above, we briefly discuss the communication cost associated with our proposed approach. In short, there is no additional communication overhead. The computational overhead associated with applying periodic optimization as part of the Ditto training framework is either, forming and solving the QP problem on a subset of training data for MK-MMD or performing a certain number of SGD steps on a modest, but not insignificant, network and set of variables with MMD-D. The focus of this work is demonstrating the utility of using these measures rather than optimizing the computational overhead. As such, there is a fair increase in training time when using these measures, but there is a lot of opportunity to reduce that overhead. The training time increases are discussed in Appendix D, Opportunities to reduce the cost have been added to this appendix as well.
>
> [1] K. Matsuda, Y. Sasaki, C. Xiao, and M. Onizuka. Benchmark for personalized federated
> learning.
>
> [2] Daoyuan Chen, Dawei Gao, Weirui Kuang, Yaliang Li, and Bolin Ding. pFL-bench: a comprehensive benchmark for personalized federated learning.
>
> [3] Yue Tan, Guodong Long, Lu Liu, Tianyi Zhou, Qinghua Lu, Jing Jiang, and Chengqi Zhang. Fedproto: Federated prototype learning across heterogeneous clients.

---

> ### Author Response · Authors · 2025-08-06
> **Highlighting Some Additional Experimental Results**
>
> As you are reviewing the discussion above, we thought it might be helpful to highlight some additional results that were shared with other reviewers in response to their questions during the rebuttal period. Specifically, we integrated MK-MMD and MMD-D into the MR-MTL algorithm, another pFL method with $\ell^2$ weight constraints, and evaluated performance across each of the datasets.  The results are summarized below.
>
> | Dataset        | MR-MTL | MR-MTL+MKMMD | MR-MTL+MMD-D |
> |----------------|--------|---------------|------------------|
> | **CIFAR (0.1)**      | 79.515 | **81.268**        | 80.307           |
> | **CIFAR (0.5)**     | 73.360 | 74.332        | **74.446**           |
> | **CIFAR (5.0)**      | 68.224 | 69.487        | **70.352**           |
> | **Synth (0.0)**  | 90.879 | 90.708        | **91.141**          |
> | **Synth (0.5)**  | 86.750 | **90.958**        | 90.337           |
> | **Fed-ISIC2019**    | **70.627** | 68.180        | 70.366           |
> | **RxRx1**          | 64.064 | 65.790        | **66.673**           |
>
> Augmenting MR-MTL with the MK-MMD or MMD-D measures improves performance on all datasets, except for Fed-ISIC2019. We believe these results better support our claim of transferability and also provide more evidence of the effectiveness of the approach in general.
>
> Thank you again for your time and efforts reviewing our work. We are sincerely grateful.

---

> > ### Comment · Reviewer_AkTQ · 2025-08-07
> >
> > I'd like to thank the authors for their detailed explanation and additional experiments. I am now leaning towards acceptance. The paper would be strengthened by including an intuitive discussion of why combining latent-space constraints with weight-based constraints creates synergy, especially in feature-heterogeneous FL.

---

> > > ### Author Response · Authors · 2025-08-07
> > >
> > > We are very appreciative of your positive comments related to our response and additional experiments. Your review has helped us make notable improvements to our work. We agree that adding a clear and intuitive discussion around the benefits of combining latent-space and weight-based constraints would strengthen this work. Such a discussion will be added to the main results in Section 6.
> > >
> > > Thank you again for your time and detailed review.

---

### Official Review · Reviewer_T2e1 · 2025-07-01

**Clarity:** 3
**Significance:** 2
**Originality:** 1
**Rating:** 4
**Confidence:** 4

**Summary:**

This paper extends the Ditto framework for personalized Federated Learning (pFL), by replacing or augmenting the way of penalizing divergence of the local model from the global model, with adaptive latent-space regularization using Maximum Mean Discrepancies (MK-MMD and MMD-D). The paper shows that using these alternative measures is beneficial in scenarios with high feature heterogeneity between clients.

**Questions:**

### Questions

1) In the paper, it is stated that the effectiveness of MMD measures is highly sensitive to the choice of kernel; however, only the RBF kernel is evaluated in the experimental section. Would it be possible to provide more insights on the choice of kernels other than RBF?

2)  ⁠Why have Ditto results not been included in Table 1? A comparison with Ditto may be beneficial to interpret results.

3) Could the authors please elaborate more on what they mean by the increasing advantages of using MMD measures in an iterative training process in the sentence in rows 178-179?

### Minor suggestions

1) The introduction should give more relevance to MMD-D and MK-MMD. Additionally, references to other algorithms in rows 39-45 may be included and better characterized in the related work section, while symbols should be introduced in the methodology section.

2) The methodology section provides an overview of the Ditto algorithm and the MMD measures, which is acceptable. However, paragraph 3.5 should provide a more thorough description of the main innovative contributions of the proposed approach, including how it could be adapted or integrated into pFL methods, as well as its advantages.

**Ethical Concerns:**

["NO or VERY MINOR ethics concerns only"]

**Final Justification:**

After revising all the reviews and the rebuttal answers provided by the authors, I am very satisfied with the answer provided and now lean towards acceptance.

**Limitations:**

yes

**Paper Formatting Concerns:**

No major formatting issues

**Quality:**

2

**Strengths And Weaknesses:**

### Strengths

1) Building upon existing literature, the paper formally and clearly characterizes the Ditto and MMD measures used to discourage divergent model weights or feature maps in pFL.

2) The proposed framework is simple and generalizable to pFL algorithms other than Ditto.

### Weaknesses

1) The paper proposes the application of existing measures, taking inspiration from Ditto. However, the use of MMD measures in FL applied to latent feature maps is not entirely new in the literature [1, 2], and the relationship between the proposed method and existing literature has not been sufficiently discussed.

2) The related works section could be improved. Specifically, an overview of techniques for pFL based on methods other than regularization techniques, and how the proposed framework positions with respect to these methods, is absent. Due to page limitations, you can also consider expanding the related works in the appendix.

3) The computational and communication costs of the proposed combinations of MMD penalty-based pFL methods have not been discussed.

4) The experiment section is not convincing. Specifically, the settings considered are overly simplistic, using too few clients and vanilla models. It would be beneficial to include experiments with other widely used networks in FL, such as ResNet20 or MobileNetV2. These networks are lightweight yet well-established in the existing literature. Additionally, increasing the number of clients, ideally to at least hundreds, would provide a more realistic representation of real-world scenarios.

[1] Yao, Xin, et al. "Federated learning with additional mechanisms on clients to reduce communication costs."

[2] Hu, Kai, et al. "FedMMD: a federated weighting algorithm considering non-IID and local model deviation."

---

> ### Author Rebuttal · Authors · 2025-07-31
>
> Thank you for your thoughtful comments about our work. We appreciate the opportunity to more deeply discuss the results and make improvements to the manuscript. Below is an itemized list of your questions and remarks. Each is followed by a discussion and description of how they were addressed in revising the paper. Thank you again for your time.
>
> **The paper proposes the application of existing measures, taking inspiration from Ditto. However, the use of MMD measures in FL applied to latent feature maps is not entirely new in the literature [1, 2], and the relationship between the proposed method and existing literature has not been sufficiently discussed.**
>
> Thank you for identifying this gap in our related work discussion. The cited work does indeed consider incorporating MMD measures within an FL framework and should have been discussed. We have added a thorough discussion to our Related Work section to better differentiate our approach. In [2], a fixed MMD measure is used to modify server-side aggregation weights, rather than influence client-side training. The approach is a global FL method and does not optimize the MMD kernel. In [1], a fixed multi-kernel MMD measure is used to constrain the feature space of client-side models during training. The objective is to learn a single global model, rather than personalized ones, and no kernel optimization is applied. As demonstrated in our ablation studies (Figure 3, right), we compare fixed MMD kernels with various bandwidths against adaptive kernel optimization. The results indicate that none of the fixed kernels match the performance of MK-MMD or MMD-D.
>
> **The related works section could be improved. Specifically, an overview of techniques for pFL based on methods other than regularization techniques, and how the proposed framework positions with respect to these methods, is absent. Due to page limitations, you can also consider expanding the related works in the appendix.**
>
> We have significantly restructured the Introduction and Related Works section to include a broader overview of existing pFL methods, including the references mentioned in the previous remark. We have also added an additional appendix section, where we discuss other pFL frameworks and how they relate to our work. Specifically, we discuss sequential and parallel-split pFL methods such as FedPer, FedRep, APFL, and FENDA-FL, prototype based methods in FedProto, and other methods such as FedBN, FedPop, and HypCluster.
>
> **The computational and communication costs of the proposed combinations of MMD penalty-based pFL methods have not been discussed.**
>
> Thank you for pointing this out. In our original submission, we did discuss computational overhead in Appendix D, but we failed to refer to that section in the main body. So it was not well-represented to readers. We now include a reference to these details in Section 7. We have also expanded the appendix to include a more thorough discussion of ways to reduce computational cost.
>
> The communication costs of the proposed approach remain identical to that of Ditto. Regardless of constraint applied, each client exchanges only the global model with the server for aggregation. In Section 3.1, we have added a brief discussion of this, as it was not clear in our original submission.
>
> **The experiment section is not convincing. Specifically, the settings considered are overly simplistic, using too few clients and vanilla models. It would be beneficial to include experiments with other widely used networks in FL, such as ResNet20 or MobileNetV2. These networks are lightweight yet well-established in the existing literature. Additionally, increasing the number of clients, ideally to at least hundreds, would provide a more realistic representation of real-world scenarios.**
>
> In this work, we have focused on cross-silo FL, which considers a small or moderate number of clients with larger training data sets. However, we did not make this clear in our introduction and have revised it. The setting of cross-silo FL is common, especially in medical or financial data settings, where clients represent a modest collection of institutions. The datasets of the FLamby benchmark are a nice example.
>
> In terms of models, we have tried to stay consistent with benchmarks and datasets where possible. The model for CIFAR-10 is drawn from the benchmark in [3], and the model for Fed-ISIC2019 matches the one used in the FLamby benchmark [4]. For RxRx1, we use ResNet-18, which is a smaller version of the ResNet-50 model used in the work that introduced the dataset [5] and is one of the models trained in [6]. We have added notes in the paper to better motivate our choice of models.
>
> We hope this discussion helps provide a better background for the experimental setup considered.
>
> **In the paper, it is stated that the effectiveness of MMD measures is highly sensitive to the choice of kernel; however, only the RBF kernel is evaluated in the experimental section. Would it be possible to provide more insights on the choice of kernels other than RBF?**
>
> This is a great question and something that we did not communicate clearly in the original text. Other kernels do exist such as Laplace or inverse multiquadric kernels [7]. Using other kinds of kernels may well improve our results. The comment around MMD sensitivity to kernel choice was meant to imply that fixing a kernel of any type can be quite sub-optimal, motivating the optimization procedures of MK-MMD and MMD-D and the periodic re-optimization procedure we propose.
>
> We have edited the paper to make this more clear and also note that other choices of kernel, beyond RBFs, are possible.
>
> **⁠Why have Ditto results not been included in Table 1? A comparison with Ditto may be beneficial to interpret results.**
>
> Thank you for the suggestion. We now include the Ditto results in Table 1.
>
> **Could the authors please elaborate more on what they mean by the increasing advantages of using MMD measures in an iterative training process in the sentence in rows 178-179?**
>
> As the training process proceeds, the distributions over the latent space induced by the global and local models evolve. While they may stabilize later in training, the optimal choice of MMD construction at one stage of training is unlikely to be optimal at another. Generally, this is true of any fixed distance measure. However, the MK-MMD and MMD-D measures can be re-optimized. We take advantage of this fact to ensure that we maintain strong measures of latent-space divergence by periodically re-optimizing the kernels based on data samples throughout the training trajectory. We have added additional discussion to make this more clear in the manuscript.
>
> **The introduction should give more relevance to MMD-D and MK-MMD. Additionally, references to other algorithms in rows 39-45 may be included and better characterized in the related work section, while symbols should be introduced in the methodology section.**
>
> Thank you for the suggestions. We have significantly reworked the Introduction and Related Work sections. We now give more weight to MMD-D and MK-MMD in the Introduction and have moved the bulk of FL algorithm discussions to the Related Work section. This section has also been expanded to incorporate additional methods. Finally, we have removed the introduction of technical symbols in these sections, as recommended.
>
> **The methodology section provides an overview of the Ditto algorithm and the MMD measures, which is acceptable. However, paragraph 3.5 should provide a more thorough description of the main innovative contributions of the proposed approach, including how it could be adapted or integrated into pFL methods, as well as its advantages.**
>
> We agree that, throughout the paper, more explicit discussion of our contributions would be helpful. We have added more clarity around our contributions to the Introduction, Related Works, and Adaptive MMD Measures in FL sections. Specifically, our contributions are:
>
> 1. Proposing the use of theoretically supported statistical distance measures as penalty constraints for dealing with heterogeneity in pFL, moving beyond previously considered paired feature-based constraints.
>
> 2. Showing that adapting the MK-MMD or MMD-D measures through periodic re-optimization provides notable performance improvements in settings with high feature heterogeneity, where existing methods, such as Ditto, under-perform.
>
> 3. Finally, experiments with either natural or controllable levels of label and feature heterogeneity highlight the strengths and weaknesses of the types of drift penalties investigated. These experiments provide a starting point for understanding when the proposed measures perform most effectively.
>
> [1] Yao, Xin, et al. Federated learning with additional mechanisms on clients to reduce communication costs.
>
> [2] Hu, Kai, et al. FedMMD: a federated weighting algorithm considering non-IID and local model deviation.
>
> [3] Daoyuan Chen,  et al.  pFL-bench: a comprehensive benchmark for personalized federated learning.
>
> [4] J. O. du Terrail,  et al. FLamby: Datasets and Benchmarks for Cross-Silo Federated Learning in Realistic Healthcare Settings.
>
> [5] Maciej Sypetkowski, et al. RxRx1: A dataset for evaluating experimental batch correction methods.
>
> [6] Haokun Chen, et al. FRAug: Tackling federated learning with non-iid features via representation augmentation.
>
> [7] Sutherland, et al.  Generative models and model criticism via optimized maximum mean discrepancy.

---

> > ### Comment · Reviewer_T2e1 · 2025-08-05
> >
> > I would like to thank the authors for their detailed and thorough rebuttal. I am very satisfied with their responses, but I have one additional comment regarding the computational and communication costs.
> >
> > I noticed Appendix D in the original submission; however, it felt somewhat brief and lacked detailed information about the costs. Instead, it mainly provided information about the GPU used and the computation time, rather than the costs themselves. When I refer to costs, I mean those expressed in big-O notation. I recommend that these costs be clearly integrated into the final version of the manuscript, even if they are the same as those for Ditto.
> >
> > Overall, I am very pleased with the provided answers, and I am now leaning towards acceptance.

---

> ### Author Response · Authors · 2025-08-06
>
> We are happy to hear that our answers and revisions were helpful. Thank you for providing additional context around cost quantification as well.
>
> Borrowing the notation from Algorithm 1, let $p$ denote the number of arithmetic operations for a single step of SGD with batch size $b$. For Ditto, the total cost, across all $N$ clients is $\mathcal{O}(N \cdot T \cdot s \cdot p)$. With both MK-MMD and MMD-D, assume we perform the respective kernel optimization procedures after every iteration of SGD for the models and have a single latent-space penalty for simplicity.
>
> For MMD-D, let $q$ be the arithmetic operations associated with a single step of SGD for the deep kernel, $\varphi$, with batch size $b$. Denote by $k$ the number of kernel optimization steps taken. Then the total cost across all clients becomes $\mathcal{O}(N \cdot T \cdot s \cdot (p + kq))$. In most cases, $p$ will be significantly larger than $kq$, as the primary models are expected to be much larger than $\varphi$.
>
> With MK-MMD, a convex quadratic program with equality constraints needs to be constructed according to the equation on Line 159. However, the cost of this construction is much smaller than the cost of solving the system. A QP solver based on interior-point methods requires $\mathcal{O}(\sqrt{d})$ iterations, each costing $\mathcal{O}(d^3)$ arithmetic operations, where $d$ is the size of the latent space [1]. Hence, the total cost across all clients is $\mathcal{O}(N \cdot T \cdot s \cdot (p + d^{3.5}))$.
>
> We will certainly clearly integrate these cost estimates into the final version of the manuscript. Thank you again for your time and don’t hesitate to let us know if you have additional questions.
>
> [1] Yinyu Ye. Interior Point Algorithms: Theory and Analysis. John Wiley & Sons, Inc., 1997

---

> > ### Comment · Reviewer_T2e1 · 2025-08-07
> >
> > Thank you for your additional clarifications. I have no further questions and confirm my score.

---

### Official Review · Reviewer_zAAU · 2025-07-03

**Clarity:** 3
**Significance:** 3
**Originality:** 3
**Rating:** 4
**Confidence:** 3

**Summary:**

This paper proposes and investigates the application of adaptive Maximum Mean Discrepancy (MMD) measures, specifically Multi-Kernel MMD (MK-MMD) and MMD-D, as regularization terms within the Ditto framework for personalized FL. The primary contribution lies in exploring these theoretically supported statistical distance measures to constrain feature representation divergence between local and global models. The paper conducts extensive experiments across various datasets, including CIFAR-10, a synthetic dataset, Fed-ISIC2019, and RxRx1, demonstrating that these MMD-based penalties, either alone or combined with the traditional Ditto penalty, yield improved model performance, particularly in settings with pronounced feature heterogeneity.

**Questions:**

1. Could you elaborate on the criteria used for selecting the optimal depth for MMD constraints, particularly for datasets other than CIFAR-10?  This is mainly because the paper states that for CIFAR-10, the optimal depth was uniformly 1. A more detailed analysis or discussion on why deeper constraints did not yield better results, and whether this observation generalizes to all datasets, would be valuable.

2. The paper mentions that for some datasets (Synthetic and RxRx1), including the Ditto penalty with MMD-based measures, degrades accuracy or hurts performance across the board. Could you provide a more in-depth explanation or theoretical intuition for this observation?

3. Regarding the computational overhead of kernel optimization procedures, the paper acknowledges this as a limitation. Could the authors suggest specific directions for future work to reduce this overhead?

4. Could the authors consider including standard deviations for the reported average performance metrics in tables, especially for key results, as this would give a better sense of the consistency of the improvements?

**Ethical Concerns:**

["NO or VERY MINOR ethics concerns only"]

**Final Justification:**

I appreciate the response, and I still believe that some of my concerns are still valid. I will not change my score.

**Limitations:**

yes

**Quality:**

3

**Strengths And Weaknesses:**

Strengths:

1. The paper presents a technically sound exploration of adaptive MMD measures in personalized FL. The methods are clearly articulated, which provides mathematical formulations for MMD, MK-MMD, and MMD-D, and details their integration into the Ditto algorithm.

2. The experimental setup is comprehensive. It utilizes both synthetic and real-world clinical datasets with varying levels of heterogeneity, which adequately tests the proposed methods.

3. The results are systematically presented and analyzed. This highlights the conditions under which the MMD-based constraints are most effective.

4. Ablation studies support the claims regarding kernel optimization and the benefits of a single constraint layer.

Weaknesses:

1- The computational overhead associated with the iterative optimization of kernels for MMD measures.

2- The paper lacks a theoretical framework that quantifies the specific conditions under which the proposed MMD measures are most effective.

3. The paper primarily focuses on the Ditto framework, and the general applicability of these MMD measures to other personalized FL methods, such as FedProx and MR-MTL, is suggested but not empirically validated in detail.

4. The experimental results are reported as the average of only three runs. This would limit the ability to fully assess the statistical significance and variability of the reported performance improvements.

---

> ### Author Rebuttal · Authors · 2025-07-31
>
> Thank you for your efforts with our manuscript and thorough review. We appreciate your comments on the technical contributions of our proposed method and comprehensive experiment setup. Below is an itemized list of your questions and remarks. Each is followed by a discussion and description of how they were addressed in revising the paper. Note that related questions or remarks may be grouped together. We look forward to additional conversations with you.
>
> **The paper lacks a theoretical framework that quantifies the specific conditions under which the proposed MMD measures are most effective.**
>
> This is accurate and a limitation that we recognize in Section 7 of the paper. While there is some theory supporting the weight-based penalty used by Ditto, theoretical characterization of the settings where MMD-based latent-space penalties become optimal is likely challenging enough to warrant a paper of its own.
>
> **The paper primarily focuses on the Ditto framework, and the general applicability of these MMD measures to other personalized FL methods, such as FedProx and MR-MTL, is suggested but not empirically validated in detail.**
>
> We have worked to expand our experimental results to better substantiate the claim that our proposed method is applicable and beneficial in other personalized federated learning algorithms. Specifically, we augmented the MR-MTL algorithm to include the MK-MMD and MMD-D constraints and evaluated such integration across all seven datasets. The results are presented below and will be added to the paper upon revision.
>
> | Dataset        | MR-MTL | MR-MTL+MKMMD | MR-MTL+MMD-D |
> |----------------|--------|---------------|------------------|
> | **CIFAR (0.1)**      | 79.515 | **81.268**        | 80.307           |
> | **CIFAR (0.5)**     | 73.360 | 74.332        | **74.446**           |
> | **CIFAR (5.0)**      | 68.224 | 69.487        | **70.352**           |
> | **Synth (0.0)**  | 90.879 | 90.708        | **91.141**          |
> | **Synth (0.5)**  | 86.750 | **90.958**        | 90.337           |
> | **Fed-ISIC2019**    | **70.627** | 68.180        | 70.366           |
> | **RxRx1**          | 64.064 | 65.790        | **66.673**           |
>
> These results demonstrate the extensibility of the proposed MMD measures to other methods. Both MK-MMD and MMD-D variants largely improve performance compared to the baseline, MR-MTL, with the exception of the Fed-ISIC2019 dataset.
>
> **Could you elaborate on the criteria used for selecting the optimal depth for MMD constraints, particularly for datasets other than CIFAR-10? This is mainly because the paper states that for CIFAR-10, the optimal depth was uniformly 1. A more detailed analysis or discussion on why deeper constraints did not yield better results, and whether this observation generalizes to all datasets, would be valuable.**
>
> Thank you for this suggestion, a deeper discussion on this has been added to Sections 5 and 6.1. CIFAR-10 served as the foundation for our first set of experiments. In observing that additional constraints at various depths did not improve performance for either MMD approach, we restricted our experiments to single layer constraints. Additional performance improvements may indeed exist for other datasets with different constraint configurations, but this simplified our search space. Our conjecture is that constraining too many layers in a relatively small model, as used in the CIFAR-10 experiments, produced too strong a constraint, hindering learning.
>
> **The paper mentions that for some datasets (Synthetic and RxRx1), including the Ditto penalty with MMD-based measures, degrades accuracy or hurts performance across the board. Could you provide a more in-depth explanation or theoretical intuition for this observation?**
>
> The Synthetic dataset is constructed to be adversarial to weight constraints, in some sense. The data points and labels are generated by transformations drawn from the same distribution, but with normally distributed variations. As such, weight-based constraints are likely to hinder learning at a certain point. The RxRx1 dataset was specifically chosen for experimentation due to the rich feature heterogeneity noted by the dataset creators, which is especially challenging for FL algorithms. We believe the proposed adaptive latent-space constraints are well suited for this heterogeneity. We have added a bit of this discussion to the revised manuscript.
>
> **The computational overhead associated with the iterative optimization of kernels for MMD measures.**
>
> **Regarding the computational overhead of kernel optimization procedures, the paper acknowledges this as a limitation. Could the authors suggest specific directions for future work to reduce this overhead?**
>
> We believe that there are several potential avenues for each method to help reduce the overhead associated with the kernel optimization procedures. The directions discussed below have been added to the revised manuscript under Appendix D.
>
> For MK-MMD, a general, python-based, quadratic program (QP) solver was used (qpth). Experimenting with more heavily optimized libraries or QP solvers specifically tailored to our system is a straightforward step. It is also possible that the systems need not be solved with significant precision, reducing the number of iterations required by the solvers. Finally, pre-conditioning techniques, such as using previous solutions as iterative starting points could reduce computation time.
>
> For MMD-D, future work could include studying utility trade-offs associated with different configurations including varying kernel update frequencies, training data sizes, and optimizers. The featurization networks are also somewhat large, consisting of four linear layers and activations with a hidden dimension of 50. Compressing or changing this architecture would likely speed up optimization.
>
> **The experimental results are reported as the average of only three runs. This would limit the ability to fully assess the statistical significance and variability of the reported performance improvements.**
>
> **Could the authors consider including standard deviations for the reported average performance metrics in tables, especially for key results, as this would give a better sense of the consistency of the improvements?**
>
> As noted in the paper, all experiments were run with three random seeds, and the reported values reflect the averages. Per your request, we now include the corresponding standard deviations for the key results presented in Table 1 and Table 2:
>
> | Dataset    | Ditto        | MMD-D        | MK-MMD -1      | MK-MMD 20      | +MMD-D        | +MK-MMD -1    | +MK-MMD 20     |
> |------------|------------------|------------------|-------------------|-------------------|-------------------|------------------|-------------------|
> | **CIFAR (0.1)**  | 84.930 ± 0.142   | 83.789 ± 0.184   | 84.136 ± 0.346   | 84.439 ± 0.304  | **85.214 ± 0.079**   | 84.723 ± 0.500  | 84.900 ± 0.317     |
> | **CIFAR (0.5)**  | 80.702 ± 0.120  | 75.094 ± 0.532  | 75.678 ± 0.000   | 76.564 ± 0.125    | 80.696 ± 0.148   | 80.936 ± 0.224  | **80.976 ± 0.365**    |
> | **CIFAR (5.0)**  | 77.658 ± 0.267  | 67.729 ± 0.591   | 68.718 ± 0.261   | 68.832 ± 0.173   | **77.739 ± 0.375**   | 77.578 ± 0.139  | **77.739 ± 0.270**   |
> | **Synth (0.0)**  | 89.129 ± 2.200  | 90.237 ± 0.405  | **91.418 ± 0.339**   | 90.066 ± 2.628   | 89.458 ± 0.347   | 89.183 ± 2.249  | 89.258 ± 2.268   |
> | **Synth (0.5)**  | 85.533 ± 0.250  | **91.270 ± 0.308**   | 91.137 ± 0.694   | 90.262 ± 0.144   | 89.695 ± 0.185   | 88.154 ± 1.816  | 88.104 ± 0.721   |
> | **Fed-ISIC2019**    | 71.350 ± 1.191   | 64.302 ± 1.546  | 60.168 ± 2.978   | 62.677 ± 0.603   | **72.260 ± 1.240**    | 71.169 ± 1.369  | 71.267 ± 1.036   |
> | **RxRx1**      | 65.629 ± 1.935  | 67.478 ± 1.199  | 65.861 ± 1.151   | 67.078 ± 0.394   | **67.755 ± 0.671**   | 65.984 ± 0.774  | 66.892 ± 0.851  |
>
> In personalized federated learning, especially under heterogeneous settings, a certain degree of performance variation is expected and has also been reported in previous work [1].
>
> In our experiments, we specifically designed the datasets to reflect different types of heterogeneity. CIFAR-10 was partitioned to emphasize label heterogeneity, while the Synthetic dataset was generated to showcase feature heterogeneity among clients. Among the real-world datasets, Fed-ISIC2019 and RxRx1 can contain both forms, though Fed-ISIC2019 primarily exhibits label heterogeneity, whereas RxRx1 is more dominated by feature heterogeneity.
>
> Our results support the hypothesis that Ditto, which applies a weight drift constraint, is particularly effective in handling label heterogeneity. However, it tends to underperform when feature heterogeneity is the dominant factor. This is evident in the observed accuracy improvements. Additionally, although less prominent, we also observe that incorporating the MMD measures on top of Ditto provides further gains in label heterogeneous settings existing in the rest of the datasets.
>
> [1] Matsuda, K., Sasaki, Y., Xiao, C., & Onizuka, M. (2023). Benchmark for personalized federated learning.

---

> > ### Comment · Reviewer_zAAU · 2025-08-05
> >
> > I thank the authors for their response; much appreciated.
> >
> > I still believe that adding a theoretical framework that quantifies the specific conditions under which the proposed MMD measures are most effective would be a good contribution to a paper, at least a discussion around it.

---

> > > ### Author Response · Authors · 2025-08-06
> > >
> > > Thank you for sharing your thoughts with us. We are also very interested in developing such a theoretical framework. There are a number of interesting technical challenges in such a construction, especially given the evolving nature of the feature spaces and the MMD measures themselves. We believe our work provides an important first step in motivating further research in this area. Should this paper move forward, we will add a discussion around potential building blocks of such theory in a section of the appendix.
> > >
> > > We hope our responses and additional experiments satisfactorily addressed all of your questions and the other weaknesses cited in your original review. Please don’t hesitate to let us know if you have further questions. Thank you again for your time.

---

### Note · Authors · 2025-08-11

We would like to take this opportunity to say a final thank you to the reviewers for all of their questions and constructive feedback. Their comments and suggestions helped us strengthen this research and improve the presentation of our ideas.

Based on the discussions with reviewers, we believe that there are no outstanding items or questions requiring further clarification from our side. The new experimental results and revisions provide additional support for the advantages of the proposed adaptive constraints in personalized federated learning (pFL). The full set of experiments demonstrate performance improvements across a number of tasks, models, and different pFL algorithms.

As noted in the discussions, we are committed to incorporating each of the revision suggestions provided by the reviewers, including additional details around computational cost, the potential complementary action of latent-space and weight-based constraints, and possible building blocks for theory characterizing when the proposed measures are optimal. Finally, we are excited about the future work motivated by our results.

Thank you for your time and consideration of our work.

---

### Decision · Program_Chairs · 2025-09-17

**Decision:**

Accept (poster)

**Comment:**

This paper explores adaptive MMD-based constraints within the Ditto framework for personalized federated learning. Reviewers agree the paper is well written, technically sound, and supported by systematic experiments and ablations. The major concerns regard the limited novelty of applying MMD in this setting, modest empirical improvements, lack of large-scale or diverse experimental validation, and incomplete discussion of costs and prior related work. The rebuttal and discussions sufficiently address the raised concerns, and all the reviewers lean toward the acceptance rating. The AC stands with the reviewers for the decision.